# Deciphering the transcriptional regulation of the catabolism of lignin-derived aromatics in *Rhodococcus opacus* PD630

Jinjin Diao[1], Rhiannon Carr [1] & Tae Seok Moon [1,2 ✉]

*Rhodococcus opacus* PD630 has considerable potential as a platform for valorizing lignin due to its innate "biological funneling" pathways. However, the transcriptional regulation of the aromatic catabolic pathways and the mechanisms controlling aromatic catabolic operons in response to different aromatic mixtures are still underexplored. Here, we identified and studied the transcription factors for aromatic degradation using GFP-based sensors and comprehensive deletion analyses. Our results demonstrate that the funneling pathways for phenol, guaiacol, 4-hydroxybenzoate, and vanillate are controlled by transcriptional activators. The two different branches of the β-ketoadipate pathway, however, are controlled by transcriptional repressors. Additionally, promoter activity assays revealed that the substrate hierarchy in *R. opacus* may be ascribed to the transcriptional cross-regulation of the individual aromatic funneling pathways. These results provide clues to clarify the molecule-level mechanisms underlying the complex regulation of aromatic catabolism, which facilitates the development of *R. opacus* as a promising chassis for valorizing lignin.

[1] Department of Energy, Environmental and Chemical Engineering, Washington University in St. Louis, St. Louis, MO 63130, USA. [2] Division of Biology and Biomedical Sciences, Washington University in St. Louis, St. Louis, MO 63130, USA. ✉email: tsmoon@wustl.edu

The serious environmental problems caused by society's dependence on fossil fuels, including climate crisis and ocean pollution, make it important to seek alternative methods to produce chemicals and fuels, particularly from renewable non-food biomass[1]. Lignocellulose, which consists of carbohydrate polymers (e.g., cellulose and hemicellulose) and aromatic polymers (e.g., lignin), represents the most abundant raw material for the potential production of next-generation renewable biofuels and chemicals[2]. To date, the major components of lignocellulose—cellulose, and hemicellulose—have been efficiently converted into various chemicals through biochemical routes[3]. Lignin, which is the second most abundant polymer on earth, holds promise as a renewable feedstock for the production of fuels and platform chemicals, due to its high carbon-to-oxygen ratio (above 2:1) and great energy density[4–7]. Moreover, recent research indicates that converting lignin to high-value fuels and chemicals would improve the overall competitiveness of biorefineries[8–12]. However, due to the structural heterogeneity of lignin, the depolymerization process typically results in diverse aromatic products, which are challenging to valorize[4,13]; consequently, lignin is still under-utilized and treated primarily as waste[14]. To date, the most predominant strategies used for valorization—including depolymerization and fragmentation—require extensive separation and purification procedures, which are commercially non-profitable due to low yields and the low quality of their final products.

Beyond chemical processing, biological treatment is a promising choice for lignin valorization; particularly, bacterial systems are increasingly attracting attention due to their inherent "biological funneling" processes[15,16]. Diverse aromatic streams can be funneled into uniform compounds (catechol (CAT) and protocatechuic acid (PCA)) and then be degraded through the β-ketoadipate pathway[17], a process that can potentially overcome the challenges associated with the heterogeneity of lignin breakdown products[18]. *Rhodococcus opacus* PD630 (hereafter, *R. opacus*), a Gram-positive soil bacterium that has the natural ability to tolerate and consume toxic aromatic compounds, has been considered a promising chassis for producing valuable products from aromatics[19–23]. Previous transcriptomic analysis of *R. opacus* cells grown on single aromatic compounds has identified the distinct funneling pathways for five lignin model compounds (benzoate, 4-hydroxybenzoate, phenol, vanillate, and guaiacol)[18,24]. Moreover, the roles of the two branches of the β-ketoadipate pathway in the degradation of those five compounds have been confirmed by gene knockout experiments[18].

Constitutively maintaining this catabolic flexibility could impose a metabolic burden on the host microbe, but this handicap is typically overcome by arranging the genes of each degradative pathway as operons whose summed expression is controlled by specific regulators and inducers. Thus, the success of a particular catabolic pathway depends not only on the efficacy of the catabolic enzymes but also on the specific regulatory elements governing their expression[25]. As genomic, genetic, and biochemical data have been accumulated, various regulatory proteins that control the expression of the aromatic degradation pathways have been identified and classified into different protein families[26]. Of additional interest are the sensory mechanisms of the regulatory elements, because regulatory proteins and their cognate promoters have the potential to be developed into biosensors that can be used in many applications, including drug discovery, biomedicine, food safety, defense, security, and environmental monitoring[27]. In *R. opacus*, comparative transcriptomics has revealed that the five aromatic funneling pathways previously identified were significantly upregulated in response to a subset of the aromatic compounds tested, suggesting that the expression of those pathway genes is likely to be tightly controlled by specific regulatory mechanisms. Furthermore, via the application of whole genome sequencing and comparative genomics, we have identified different families of transcription factors (TFs) that are located adjacent to *R. opacus*'s proposed aromatic catabolic pathways[18,24]. However, the specific roles of these TFs in regulating the aromatic degradation pathways—such as the signals that trigger pathway expression and the exact mechanisms of activation and/or repression—are still unclear.

In natural environments, carbon sources are commonly found as heterogeneous mixtures. To handle these mixtures, including the portion of aromatic compounds that are toxic, most bacteria have evolved a hierarchy of substrate utilization that enables them to quickly adapt their intracellular metabolic network toward a preferred substrate, which is vital for competition in these environments. This phenomenon, termed carbon catabolite repression (CCR), has been extensively reported in the utilization of sugar mixtures and non-sugar substrates[28–30], but the number of studies conducted on substrate combinations containing only aromatic compounds is limited. To date, most studies have focused on benzoate and 4-hydroxybenzoate, which are commonly metabolized via the two parallel branches of the β-ketoadipate pathway. For example, in the γ-proteobacteria *Pseudomonas putida* PRS2000 and *Acinetobacter* sp. Strain ADP1, benzoate has been found to be the preferred substrate of the two[31,32]. Similarly, in the β-proteobacterium *Cupriavidus necator* JMP134, the same utilization hierarchy between benzoate and 4-hydroxybenzoate has been observed[33]. In *Rhodococcus* sp. strain DK17, a catabolite repression-like response has been reported when cells are simultaneously provided with benzoate and phthalate[34]. Moreover, benzoate catabolite repression of phenol degradation has been observed in *Acinetobacter calcoaceticus* PHEA-2[35]. Finally, in *R. opacus*, in a mixture of the five lignin model compounds previously described, benzoate was found to be consumed preferentially[18], which suggests the existence of a substrate hierarchy. Although it is important for microbial lignin conversion strategies, this sequential consumption order is still underexplored.

In this study, to identify those TFs involved in regulating the degradation of lignin model compounds, we selected potential TFs in the genomic neighborhood of aromatic operons and knocked them out via homologous recombination. By using metabolite sensors derived from native *R. opacus* promoters that can detect aromatic compounds and by comparing the cell growth and aromatic consumptions of these TF deletion mutants to those of the wild type (WT), we evaluated the roles of the candidate TFs in regulating the degradation pathway of each lignin model compound. To establish the substrate hierarchy of the tested lignin model compounds, we performed time-course analyses of the consumption of individual aromatics in the mixture, revealing that these compounds were consumed in a distinct order. Moreover, by testing the responses of the funneling pathways in WT cells grown on both individual lignin model compounds and a mixture, we confirmed that *R. opacus* can differentially and specifically regulate the funneling pathways in response to specific compounds, which is critical for the utilization of the preferred aromatic substrate. Taken together, these results advance our understanding of the regulatory patterns of the aromatic degradation pathways, which is critical to constructing a more efficient bacterial chassis for comprehensively utilizing lignin. Moreover, these insights into the mechanism of hierarchical utilization of aromatics in *R. opacus* are of great significance for achieving rapid consumption of complex aromatic mixtures, enabling more cost-effective conversion of lignin into fuels, chemicals, and materials.

## Results

**Transcriptional regulation of the funneling pathways responsible for the degradation of aromatics via the CAT branch of the β-ketoadipate pathway**. Previously, we found the ΔcatB (cis,cis-muconate cycloisomerase knockout) strain was unable to grow using phenol, guaiacol, and benzoate as sole carbon sources, demonstrating that those three compounds are metabolized through the CAT branch of the β-ketoadipate pathway in *R. opacus*[18]. Additionally, by using RNA-Seq, several funneling pathway clusters, including those for phenol, guaiacol, and benzoate, were identified[18]. Because those genes were found to be upregulated by *R. opacus* in the presence of the relevant compounds, it was hypothesized that the induction of the funneling pathways is controlled by specific regulatory mechanisms. To identify the respective protein regulators, potential TFs, encoded by genes adjacent to the catabolic clusters, were selected (Supplementary Table 1).

For phenol degradation, a pair of similar phenol hydroxylase clusters (designated *pheB1A1* and *pheB2A2*) were identified in the *R. opacus* genome, and these genes were significantly upregulated in the presence of phenol[18,24]. Furthermore, two AraC-type TFs, *pheR1* (LPD06739) and *pheR2* (LPD06574), which are located adjacent to these clusters but transcribed divergently (Fig. 1a, b), were selected for examination. To confirm the transcriptional regulation of these two clusters, transcriptional fusions of the respective promoters (P*pheB1* and P*pheB2*) with the *gfp+* reporter gene were constructed, and expressed in the WT strain (Fig. 1c, f). For P*pheB1*, the promoter activity was markedly increased with cultivation on phenol, but not on *cis-cis* muconate, 4-hydroxybenzoate, or PCA. In comparison, significant induction of P*pheB2* was observed with cultivation on both phenol and *cis-cis* muconate (Fig. 1d, g and Supplementary Fig. 1).

To elucidate the role of the two adjacent TFs in regulating the expression of these phenol funneling pathways, we generated two TF deletion mutants (Δ*pheR1* and Δ*pheR2*) and measured the promoter activity of P*pheB1* and P*pheB2*, respectively, using the previously-described *gfp+* constructs. In Δ*pheR1*, expression from promoter P*pheB1* was severely inhibited but remained detectably ON (Fig. 1d). In Δ*pheR2*, however, the fluorescence output of P*pheB2* was completely OFF (Fig. 1g). These results indicate that *pheR1* and *pheR2* act as activators of their respective phenol degradation clusters (Fig. 1c, f). Having established their roles, we analyzed cell growth and phenol consumption in the two mutants. When fed phenol as the sole carbon source, no cell growth or phenol consumption was observed in the Δ*pheR2* strain; in contrast, cell growth and phenol consumption of Δ*pheR1* strain were comparable to those of the WT strain (Fig. 1e, h). Cross-validation showed that knocking out either *pheR1* or *pheR2* had no effect on the induction of the non-cognate hydroxylase cluster (Fig. 1i, j). Additionally, we found that in a dual mutant (Δ*pheR1* Δ*pheR2*), the fluorescence output of the promoter P*pheB1* was null, suggesting that this promoter may have crosstalk with TF *pheR2* (Fig. 1k).

In *R. opacus*, although previous studies have shown that the genes encoding the phenol hydroxylase in these two clusters have a high identity to each other, and that this strain appears to utilize both copies of the two genes[24], our results led us to hypothesize that only the *pheB2A2* copy is essential for phenol degradation. To test this hypothesis, we generated a *pheB1* deletion mutant. As expected, when fed phenol as the sole carbon source, the WT and Δ*pheB1* strains showed statistically identical cell growth. Repeated attempts to disrupt the *pheB2* failed, so a T7 RNAP-based CRISPRi platform[36] was applied to test this putatively essential phenol degradation cluster (*pheB2A2*). In the strain expressing sgRNA PHE_1 (targeting *pheB2*), no significant increase in OD$_{600}$ was observed under the induced condition, but growth was comparable to that of the control strain when uninduced

(Supplementary Fig. 2). This result confirmed that *pheB2A2* is the pivotal cluster for phenol degradation in *R. opacus*.

To identify the potential TF involved in regulating the guaiacol degradation pathway, a putative AraC-type TF *guaR* (LPD06577) —which is located adjacent to the deduced guaiacol funneling cluster—was selected as a knockout target (Fig. 2a)[18]. Specifically, the construct P*cyp255*-GFP+ was expressed in both WT and Δ*guaR* strains (Fig. 2b). In the WT strain, the promoter activity was remarkably increased when cells were cultivated on guaiacol, but not on CAT, *cis-cis* muconate, 4-hydroxybenzoate, or PCA (Fig. 2c and Supplementary Fig. 1c). In the mutant Δ*guaR*, no expression from the promoter was observed with cultivation on guaiacol (Fig. 2c), indicating that *guaR* works as an activator with guaiacol as the inducer (Fig. 2b and Supplementary Fig. 1c). Furthermore, when the mutant was fed guaiacol as the sole carbon source, neither cell growth nor guaiacol consumption was detected (Fig. 2d), suggesting that *guaR* is the essential regulator for guaiacol degradation.

A putative funneling pathway for benzoate was proposed from the transcriptomic analysis[18]. By taking a close look at the genome architecture, we found that this cluster shares the AraC-type TF *guaR* (LPD06577) with the guaiacol funneling pathway (Fig. 2e). To analyze the potential role of *guaR* in regulating benzoate funneling pathway, the construct P*benA*-GFP+ was expressed in both WT and Δ*guaR* strains, and the fluorescence output was measured in response to benzoate. The results showed that expression from this promoter decreased ~31% when knocking out *guaR*, but that the mutant maintained an identical benzoate consumption rate to that of the WT strain (Fig. 2f, g). These results indicate that *guaR* may not be the direct regulator of benzoate degradation.

**Transcriptional regulation of the funneling pathways responsible for the degradation of aromatics via the PCA branch of the β-ketoadipate pathway**. In addition to the CAT branch of the β-ketoadipate pathway, the PCA branch has been identified as playing a role in the degradation of aromatics, including 4-hydroxybenzoate and vanillate. Similar to phenol, guaiacol, and benzoate in the CAT branch, the funneling pathways for these two compounds have been proposed based on transcriptomic analysis[18]. For 4-hydroxybenzoate, a putative 4-hydroxybenzoate monooxygenase *pobA* (LPD06764) was identified, which is responsible for the conversion of 4-hydroxybenzoate to protocatechuate. Adjacent to this gene are two proposed TFs, IclR-type regulator *hbaR1* (LPD06765) and Tet/AcrR-type regulator *hbaR2* (LPD06763) (Fig. 3a, e). To study the roles of both TFs in regulating the 4-hydroxybenzoate degradation pathway, the construct P*pobA*-GFP+ was expressed in both WT and the TF deletion mutant strains (Δ*hbaR1* and Δ *hbaR2*). In the WT strain, the promoter activity was markedly increased during cultivation on 4-hydroxybenzoate, but not on PCA or on CAT (Supplementary Fig. 1e). In the mutant Δ*hbaR1*, no transcription from the promoter was observed, but in Δ*hbaR2*, the fluorescence output of the promoter was identical to that of the WT strain (Fig. 3c, f). These results demonstrate that *hbaR1* acts as an activator in regulating the 4-hydroxybenzoate funneling pathway with 4-hydroxybenzoate as the inducer (Fig. 3b). We also tested the two knockout mutants by feeding them with 4-hydroxybenzoate as the sole carbon source. No cell growth or 4-hydroxybenzoate consumption was observed in Δ*hbaR1*, but in Δ*hbaR2*, cell growth and 4-hydroxybenzoate consumption were comparable to those of the WT strain (Fig. 3d, g). These results indicate that *hbaR1* is necessary for the regulation of 4-hydroxybenzoate funneling pathway in *R. opacus*.

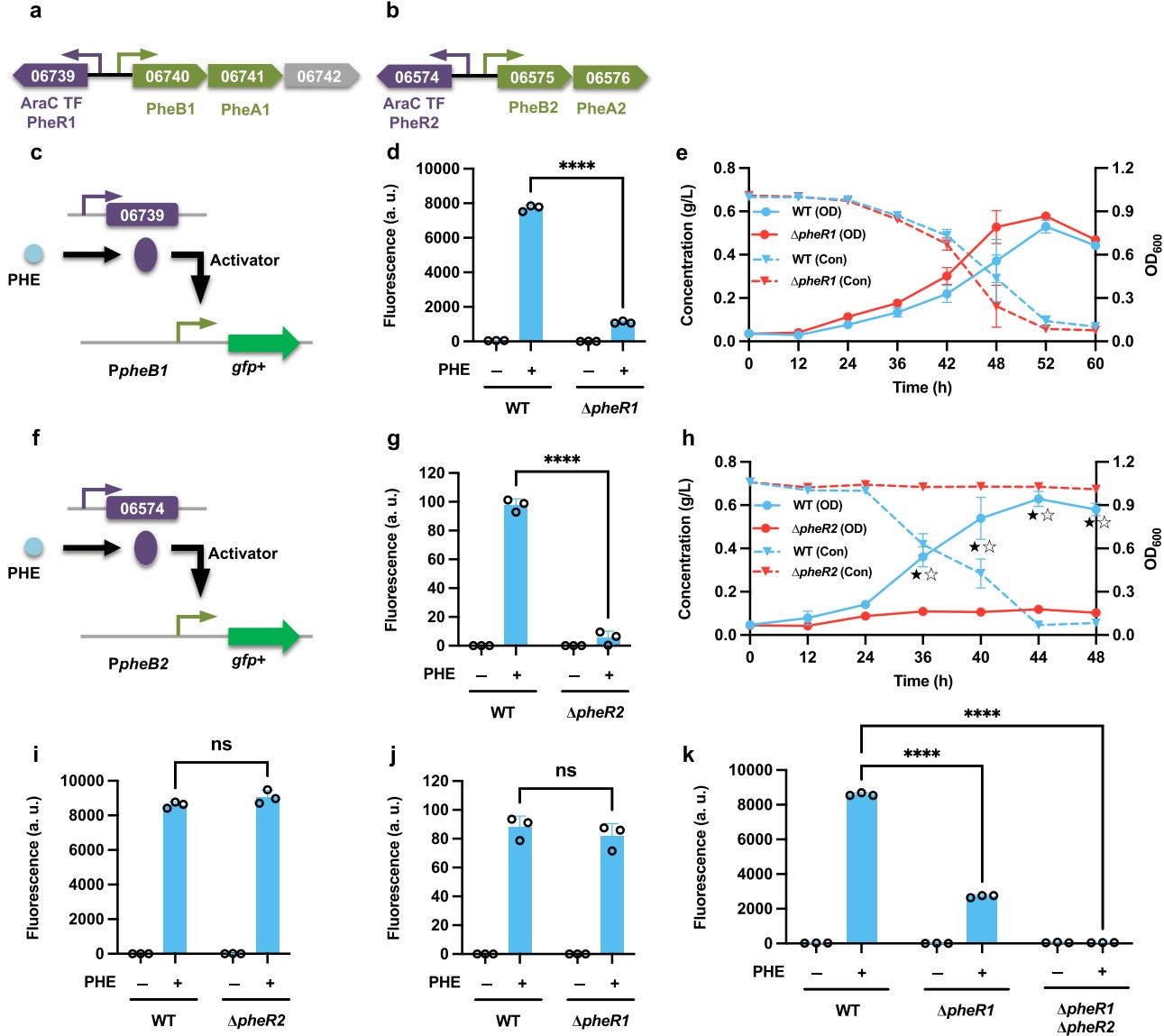

**Fig. 1 Functional analysis of the phenol-responsive transcription factors *pheR1* (LPD06739) and *pheR2* (LPD06574).** The *R. opacus* PD630 genome contains two phenol degradation operons, depicted in **a**, **b**; the schematics include the promoters (small arrows) and proposed corresponding transcription factors (TFs), with operon genes indicated by LPD gene numbers corresponding to the NCBI database (Refseq, CP003949.1). For the full annotations of the pathway genes, see Supplementary Table 1. **c**, **f** To analyze the function of the proposed transcription factors *pheR1* and *pheR2* in regulating these operons, the transcriptional constructs P*pheB1*-GFP+ and P*pheB2*-GFP+ were expressed in WT and the corresponding TF deletion mutant strains. **d**, **g** The normalized fluorescence of the WT and Δ*pheR1* and Δ*pheR2* was measured with and without supplemental phenol (PHE) (****, $P < 0.0001$, unpaired two-tailed *t* test). **e**, **h** Comparisons of cell growth (OD) and phenol consumption (Con) between WT and the two TF deletion mutant strains Δ*pheR1* and Δ*pheR2* when fed 0.7 g/L PHE as the sole carbon source; cell growth and phenol consumption were significantly reduced in the Δ*pheR2* strain (★, $P < 0.05$ for OD; ☆, $P < 0.05$ for Con; two-tailed mixed model ANOVA with Sidak's multiple comparisons). **i**, **j** The transcriptional activity of the promoter P*pheB1* in the mutant Δ*pheR2* and the transcriptional activity of the promoter P*pheB2* in the mutant Δ*pheR1*. Measurements were conducted with and without supplemental PHE (ns, not significant, unpaired two-tailed *t* test). **k** Comparison of the transcriptional activity of the promoter P*pheB1* in the different knockout strains when treated with and without supplemental PHE (****, $P < 0.0001$, unpaired two-tailed *t* test). For all fluorescence assays, cell cultures contained 1 g/L glucose with (+) or without (−) 0.3 g/L PHE as the carbon source; all the fluorescence values were determined in the early stationary phase and normalized to an optical density at 600 nm (OD$_{600}$). All values represent the mean of triplicate cultures, with error bars depicting the standard deviation from that mean.

Similar to the 4-hydroxybenzoate funneling pathway, two putative TFs, IclR-type regulator *vanR1* (LPD00562) and RrfR-type regulator *vanR2* (LPD00561), are found adjacent to the vanillate funneling pathway (Fig. 4a, e). To elucidate the roles of these TFs in regulating the vanillate degradation pathway, the construct P*vanA*-GFP+ was expressed in both WT and the corresponding TF deletion mutants (Fig. 4b). In the WT strain, the promoter demonstrated a remarkably strong response to vanillate, but not to PCA or CAT (Supplementary Fig. 1f). In Δ*vanR1*, the fluorescence output of P*vanA* was completely OFF and no cell growth or vanillate consumption was observed when cells were fed with vanillate as the sole carbon source (Fig. 4c, d). Knocking out *vanR2* had the unexpected effect of significantly decreasing the promoter activity of P*vanA*, as well as limiting VAN consumption (Fig. 4f, g). When examining the genome architecture more closely, we noted that these two TFs have a

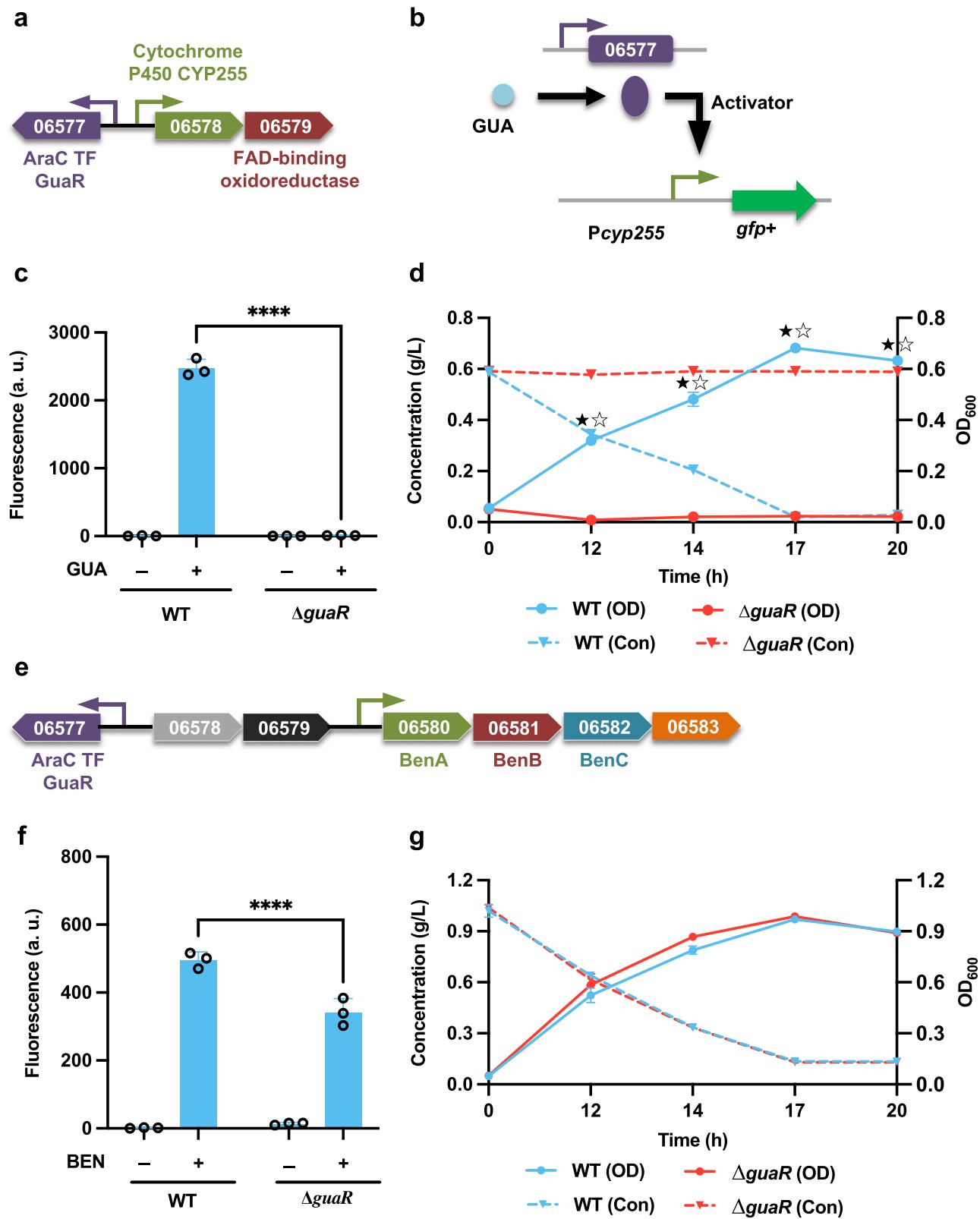

short overlap region in the C terminus. The phenotype of Δ*vanR2*, which was created with *vanR1* intact, is characterized in part by the negative effect on the expression of *vanR1*, an effect which may be explained by the gene deletion causing unexpected context effects on neighboring genes[36]. Taken together, these findings suggest that *vanR1* is a key activator that regulates the vanillate funneling pathway in *R. opacus*.

**Transcriptional regulation of the β-ketoadipate pathway.** The funneling pathways described above convert aromatic compounds into intermediate compounds, CAT, and protocatechuate for subsequent degradation by the β-ketoadipate pathway. Based on the previous transcriptomic analysis, three potential β-ketoadipate gene clusters are upregulated in WT when treated with aromatic compounds; furthermore, the fundamental roles of

**Fig. 2 Functional analysis of the AraC-type transcription factor GuaR (LPD06577). a, e** The guaiacol (GUA) and benzoate (BEN) degradation operon with an annotated transcription factor (TF). Promoters are represented as small arrows. Genes are shown with LPD gene numbers from the NCBI database (Refseq, CP003949.1). For the full annotations of the pathway genes, see Supplementary Table 1. **b** The proposed model for the role of *guaR*. **c, f** To analyze the function of the proposed TF, the constructs P*cyp255*-GFP+ and P*benA*-GFP+ were expressed in WT and the corresponding TF deletion mutant strains; the normalized fluorescence of the WT and Δ*guaR* was measured with and without supplemental GUA (**c**) or BEN (**f**). **d, g** Comparisons of cell growth (OD) and aromatic consumption (Con) between the WT and Δ*guaR* strains when fed 0.6 g/L GUA (**d**) or 1.0 g/L BEN (**g**) as the sole carbon source. For all fluorescence assays, cell cultures contained 1 g/L glucose with (+) or without (−) 0.3 g/L GUA or 1.0 g/L BEN as the carbon source; all the fluorescence values were determined in the early stationary phase and normalized to $OD_{600}$. Unpaired two-tailed *t* test was used to compare the variation in the change of fluorescence of the mutants treated with respective aromatic against that of the WT control (****, $P < 0.0001$); for cell growth and aromatic consumption, the variations in the changes of cell density (OD) and aromatic concentration (Con) of the mutant were compared against those of the WT control (★, $P < 0.05$ for OD; ☆, $P < 0.05$ for Con; two-tailed mixed model ANOVA with Sidak's multiple comparisons). All values represent the mean of triplicate cultures, with error bars depicting the standard deviation from that mean.

the CAT and PCA branches of the β-ketoadipate pathway have been confirmed by gene knockout experiments[18]. For the CAT branch, the critical genes and an IclR-type TF *catR* (LPD06569) are clustered but transcribed divergently (Fig. 5a). To analyze the transcriptional regulation of this cluster, we first localized the promoters P*catA* and P*catR* by comparative analysis[37]. Proposed transcription start sites (TSSs) were found 80 nt and 127 nt upstream of the translation start sites of *catA* and *catR*, respectively. Additionally, the −35 and -10-like elements of the promoter regions were found with appropriate spacing within the sequences upstream of the TSSs (Fig. 5g). Moreover, for the promoter P*catR*, the supposed -35 and -10 elements overlap with the putative IclR-type regulator binding site, indicating a self-regulating mechanism for the expression of this TF.

To study the roles of *catR* in regulating this cluster, we expressed transcriptional constructs P*catA*-GFP+ and P*catR*-GFP+ in both the WT and corresponding TF deletion mutant strains (Fig. 5b, c). In the WT strain, the promoter P*catA* was responsive to benzoate, phenol, guaiacol, and CAT, but not to PCA (Supplementary Fig. 1g and 3b–d). In Δ*catR*, the activity of P*catA* was strongly increased with the cultivation of both glucose and the respective aromatic compounds (Supplementary Fig. 3b–d), indicating that *catR* acts as a repressor (Fig. 5b). Moreover, as expected, knocking out *catR* modestly enhances the consumption of all three aromatic compounds (Supplementary Fig. 3e–g). This promoter is responsive when cultivated on phenol, benzoate, or guaiacol, so we hypothesized that the real effector compound might be one of the intermediates of the CAT degradation pathway. To confirm this hypothesis, we first tested the WT and TF deletion mutant strains with cultivation on CAT. Analysis showed that P*catA* displayed only low basal activity in the glucose condition, but that the promoter activity was markedly enhanced in CAT (Fig. 5d). Surprisingly, the same trend of promoter activity was observed when cells were grown on *cis-cis* muconate (Fig. 5e). As it is known that CAT can be spontaneously converted into *cis-cis* muconate by the basal expression of the CAT 1,2-dioxygenase (*catA*), this finding suggests that the real effector compound may be *cis-cis* muconate.

A previous study has revealed that in *Rhodococcus erythropolis*, expression of the TF responsible for regulating the CAT degradation cluster is regulated by auto-regulatory repression[37]. To examine whether the regulation pattern of *catR* (LPD06569) in *R. opacus* is comparable, the promoter activity of P*catR* was examined with cultivation on various carbon sources. In the WT strain, promoter P*catR* showed constitutive activity when glucose was present, but in Δ*catR*, only basal expression from this promoter was observed under the same condition. Positing that ligand binding could change the conformation of *catR*, leading to the detachment of protein from the cognate promoter, we hypothesized that with ligand present in the media, the promoter activity of P*catR* would be decreased. As expected, with the

addition of *cis-cis* muconate, the output of this promoter was significantly decreased (Fig. 5f). Taken together, these results indicate that when regulating its own expression, *catR* acts as an activator in the absence of the ligand *cis-cis* muconate (Fig. 5c, h). The same assay was repeated with an array of aromatic compounds; promoter P*catR* displayed reduced activity in all cases except when cultivated on PCA, which is not converted into *cis-cis* muconate (Supplementary Fig. 1h). Moreover, to test whether *catR* displays crosstalk between the benzoate, phenol, and guaiacol funneling pathways, the constructs P*benA*-GFP+ (benzoate responsive), P*pheB2*-GFP + (phenol responsive), and P*cyp255*-GFP+ (guaiacol responsive) were expressed in WT and the mutant Δ*catR*. In the absence of *catR*, the promoter activities of P*benA* and P*pheB2* were identical to that of the WT strain; however, the output of the promoter P*cyp255* was significantly decreased in Δ*catR*, implying that *catR* may work as a weak co-activator in regulating the guaiacol funneling pathway (Supplementary Fig. 4). More intriguingly, no putative binding sites for *catR* were identified in P*cyp255*, suggesting *catR* may play an indirect role in regulating this funneling pathway.

In addition to the role of the CAT branch of the β-ketoadipate pathway in degrading phenol, guaiacol, and benzoate, the PCA branch has been identified as the key pathway for the degradation of 4-hydroxybenzoate and vanillate in *R. opacus*. Moreover, the putative gene cluster has been confirmed by gene knockout experiments[18]. In contrast to the related mycobacterium *Corynebacterium glutamicum*[38], only one IclR-type TF *pcaR* (LPD05454) is located in the PCA operon. To investigate how this single TF regulates the entire PCA operon, P*pcaH*-GFP+ and P*pcaI*-GFP+ constructs were expressed in both WT and the corresponding TF deletion mutant strains. For the orientation corresponding to coding region *pcaH-pcaF* (LPD05450-05455), promoter P*pcaH* was responsive to all the compounds we tested (Supplementary Fig. 1i), suggesting that—similar to observations in the CAT degradation pathway—the real inducer is also one of the intermediate compounds in the PCA degradation pathway. In the Δ*pcaR* strain, the promoter activity of P*pcaH* was significantly higher during cultivation on glucose, suggesting that *pcaR* may work as a transcriptional repressor (Fig. 6a-c). For the *pcaI-pcaJ* coding region (LPD05448-05449), in the presence of *pcaR*, carbon source did not make a significant difference in the promoter activity of P*pcaI*. However, in the mutant Δ*pcaR*, the promoter activity was dramatically enhanced on glucose, compared to that of WT, indicating that in regulating this coding region, *pcaR* works ligand-free as a repressor (Fig. 6d, e).

**Transcriptional cross-regulation of the funneling pathways during growth on alternative substrates.** Bacteria often degrade mixed compounds sequentially; this type of ordered catabolism, termed CCR, has been noted in many different strains[32,39–41]. In

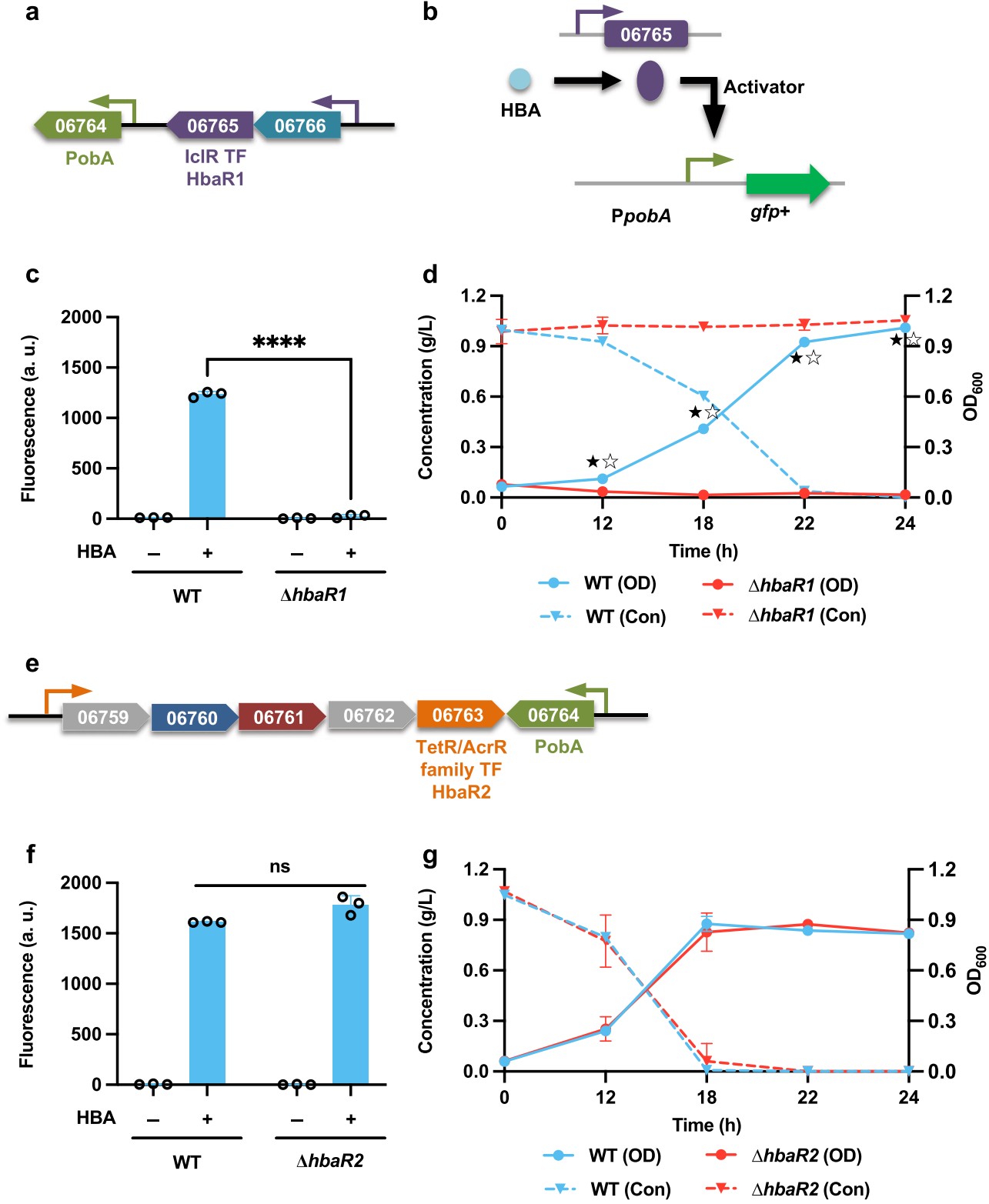

*R. opacus* specifically, a targeted metabolomic approach was applied to establish the time-resolved concentrations of each individual aromatic compound in a defined mixture, showing that this strain preferred benzoate and 4-hydroxybenzoate over phenol, guaiacol, and vanillate. In the same work, a transcriptomic study revealed that *R. opacus* can differentially and specifically regulate aromatic utilization pathways in response to aromatic mixtures[18], which led us to investigate the adaptability of *R.*

*opacus* to the presence of multiple aromatic hydrocarbons as carbon sources.

As an initial step in studying the response of *R. opacus* to simultaneously available aromatic mixtures, the effects of 4-hydroxybenzoate and benzoate on the other three model compounds' catabolism were examined. Time-course analyses of cell growth and the aromatic concentrations were conducted, and the results showed that the presence of 4-hydroxybenzoate

**Fig. 3 The transcription factor *hbaR1* (LPD06765) activates the degradation of 4-hydroxybenzoate. a, e** Two potential transcription factors, *hbaR1* (LPD06765) and *hbaR2* (LPD06763), are located near 4-hydroxybenzoate hydroxylase *pobA* (LPD06764). Promoters are represented as small arrows, and genes are shown with LPD gene numbers from the NCBI database (Refseq, CP003949.1). For the full annotations of the pathway genes, see Supplementary Table 1. **b** The proposed working model for *hbaR1*. **c, f** Analysis of the function of the proposed transcription factors. The construct P*pobA*-GFP+ was expressed in the WT and the corresponding TF deletion mutant strains. The normalized fluorescence of the WT and Δ*hbaR1* and Δ*hbaR2* was measured with and without supplemental 4-hydroxybenzoate (HBA). **d, g** Comparisons of cell growth (OD) and HBA consumption (Con) between WT and the two TF deletion mutant strains Δ*hbaR1* and Δ*hbaR2* when fed 1.0 g/L HBA as the sole carbon source. For all fluorescence assays, cell cultures contained 1 g/L glucose with (+) or without (−) 1.0 g/L HBA as the carbon source; all the fluorescence values were determined in the early stationary phase and normalized to $OD_{600}$. Unpaired two-tailed $t$ test was used to compare the variation in the change of fluorescence of the mutants treated with HBA against that of the WT control (****, $P < 0.0001$; ns, not significant). To assess cell growth and aromatic consumption, the variations in the changes of cell density (OD) and aromatic concentration (Con) of the mutants were compared against those of the WT control (★, $P < 0.05$ for OD; ☆, $P < 0.05$ for Con; two-tailed mixed model ANOVA with Sidak's multiple comparisons). All values represent the mean of triplicate cultures, with error bars depicting the standard deviation from that mean.

inhibited the catabolism of phenol and guaiacol, but not vanillate (Fig. 7a-c). Even though the utilization of 4-hydroxybenzoate was preferential, diauxic growth was not observed (Fig. 7a, c). Because the funneling pathways play important roles in the degradation of aromatics, we hypothesized that 4-hydroxybenzoate might mediate a certain form of transcriptional repression over the funneling pathway of phenol or guaiacol, in particular by inhibiting the activity of the cognate promoters. To test our hypothesis, different strains harboring the constructs P*pheB2*-GFP+ (phenol responsive), P*vanA*-GFP+ (vanillate responsive), or P*cyp255*-GFP+ (guaiacol responsive) were selected to test the activities of each promoter when treated with different combinations of aromatic mixtures (Fig. 7d–f). For phenol, transcription from P*pheB2* was significantly decreased by the presence of 4-hydroxybenzoate for the first 16 h of cultivation (Fig. 7g). Promoter P*pheB1*, which drives the expression of non-essential phenol degradation cluster was also selected and tested. Similar to the findings for P*pheB2*, the data showed that the presence of 4-hydroxybenzoate precipitates a significant delay in the induction of this promoter by phenol, and that expression increased in the latter stages of cultivation, suggesting that the transcriptional repression observed earlier may be eased as 4-hydroxybenzoate is preferentially consumed (Supplementary Fig. 5a, b). For the guaiacol-reactive promoter, P*cyp255*, a similar trend was observed (Fig. 7i). In contrast, the addition of 4-hydroxybenzoate did not cause a significant delay in vanillate induction, except at the 16 and 20 h time points (Fig. 7h).

The effect of benzoate on the catabolism of phenol, vanillate, and guaiacol was also analyzed; the results indicate that while *R. opacus* preferentially consumes benzoate over phenol and vanillate (Fig. 8a, b), no such preference was observed between benzoate and guaiacol (Fig. 8c). To test whether the similarities between 4-hydroxybenzoate and benzoate preferential consumption extended to transcriptional repression of parallel funneling pathways, the activities of the cognate promoters were examined by feeding cultures with different combinations of aromatic monomers (Fig. 8d–f). For phenol and vanillate, transcription from the cognate promoter decreased with the addition of benzoate (Fig. 8g, h and Supplementary Fig. 5c), which is consistent with the consumption profile data. Intriguingly, even though *R. opacus* could simultaneously consume benzoate and guaiacol, significant transcriptional repression of promoter P*cyp255* was observed when benzoate was available during the cultivation (Fig. 8i). Similarly, we compared the consumption profiles of benzoate and 4-hydroxybenzoate. We observed that in the WT strain, the consumption of benzoate is faster than that of 4-hydroxybenzoate (Supplementary Fig. 6a), and further observed that, in the presence of benzoate, the induction of the 4-hydroxybenzoate funneling pathway was delayed (Supplementary Fig. 6b, c).

To further specify the sequential prioritization of vanillate, phenol, and guaiacol, we conducted a combinatorial examination of the consumption profiles. For phenol+vanillate, the consumption data showed that *R. opacus* could simultaneously consume vanillate and phenol, and that VAN induction was not repressed by the addition of phenol (Supplementary Fig. 7a, b, d). In contrast, the activity of the phenol-responsive promoter P*pheB1* was significantly decreased when vanillate was available (Supplementary Fig. 7b, c). For the combinations of phenol+guaiacol and vanillate+guaiacol, we observed that *R. opacus* consumed phenol and vanillate prior to guaiacol (Supplementary Figs. 8a and 9a). As this delay corresponded to a delay in guaiacol induction in the presence of vanillate or phenol (Supplementary Figs. 8d and 9d), we posited that vanillate and phenol may inhibit the consumption of guaiacol.

The above results suggest that the sequential consumption of aromatics in *R. opacus* may be mediated through a certain form of transcriptional repression over the respective funneling pathways. To confirm this possibility, we examined the promoter activity of P*cyp255* in different mutants with cultivation on guaiacol plus 4-hydroxybenzoate or the intermediate compound PCA (4-hydroxybenzoate degradation through the PCA branch and the corresponding transcriptional regulation pattern being clearly demonstrated; Fig. 9a). In the WT, guaiacol induction was significantly inhibited when 4-hydroxybenzoate was available as additional carbon source, but with the addition of an equal amount of PCA, the induction pattern of the promoter P*cyp255* reverted to that of the guaiacol-only treatment (Fig. 9b). Similar repression was observed in the PCA scenario when additional 4-hydroxybenzoate was added (Fig. 9b), suggesting that, rather than the intermediates of 4-hydroxybenzoate degradation, 4-hydroxybenzoate itself precipitates the observed transcriptional repression. Because the TF-based "induction prevention" mechanism has been identified to describe the preferential consumption of aromatics in bacteria[31,33], to better address the possibility of 4-hydroxybenzoate repression on guaiacol utilization, we generated mutant strains defective in the catabolism of 4-hydroxybenzoate. First, the TF *hbaR1* (LPD06765)—a transcriptional activator responsible for regulating the expression of 4-hydroxybenzoate monooxygenase *pobA* (LPD06764)—was knocked out. The fusion construct P*cyp255*-GFP+ was expressed in this Δ*hbaR1* mutant, and the resulting fluorescence data indicated a lack of transcriptional repression, regardless of whether cells were cultivated with or without 4-hydroxybenzoate (Fig. 9c). Next, we deleted *pobA* (LPD06764). In this mutant, the conversion of 4-hydroxybenzoate to PCA is abolished, whereas TF *hbaR1* is still active. Monitoring the promoter activity when cells were cultivated on different aromatic combinations showed no transcriptional repression in Δ*pobA* strain (Fig. 9d), suggesting that the transcriptional repression cannot attribute to

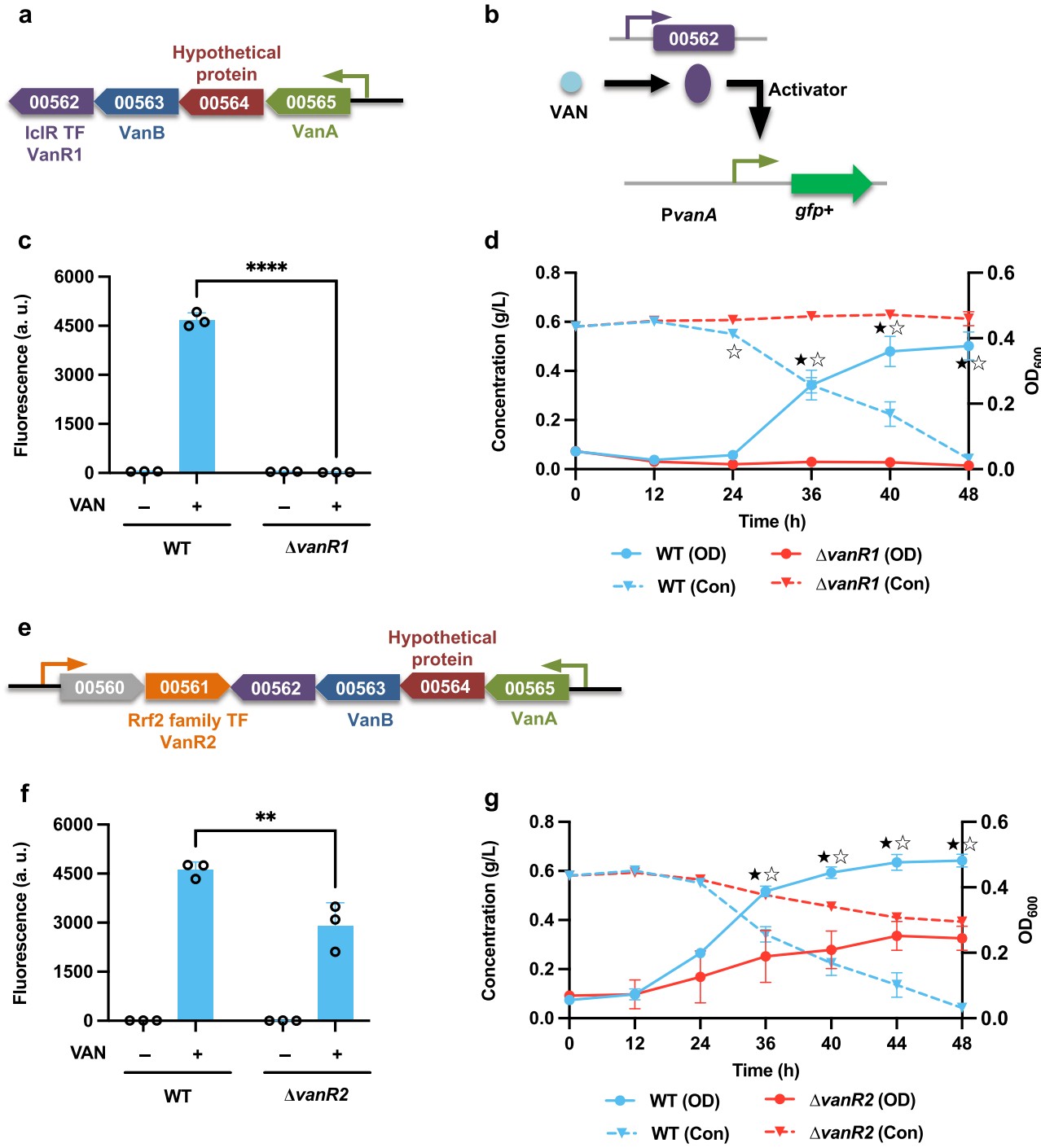

**Fig. 4 The transcription factor *vanR1* (LPD00562) activates the degradation of vanillate. a**, **e** The vanillate degradation operon in *R. opacus* is adjacent to two transcription factors, *vanR1* (LPD00562) and *vanR2* (LPD00561). Promoters are represented as small arrows, and genes are shown with LPD gene numbers from the NCBI database (Refseq, CP003949.1). For the full annotations of the pathway genes, see Supplementary Table 1. **b** The proposed working model for *vanR1*. **c**, **f** Analysis of the function of the proposed transcription factors. The construct P*van*A-GFP+ was expressed in WT and in the corresponding TF deletion mutant strains. The normalized fluorescence of the WT, Δ*vanR1*, and Δ*vanR2* was measured with and without supplemental vanillate (VAN). **d**, **g** Comparison of the cell growth (OD) and vanillate consumption (Con) between the WT and the two TF deletion mutant strains Δ*vanR1* and Δ*vanR2* when fed 0.6 g/L VAN as the sole carbon source. For all fluorescence assays, cell cultures contained 1 g/L glucose with (+) or without (−) 0.3 g/L VAN as the carbon source; all the fluorescence values were determined in the early stationary phase and normalized to OD$_{600}$. Unpaired two-tailed *t* test was used to compare the variation in the change of fluorescence of the mutants treated with VAN against that of the WT control (**, $P < 0.0021$; ****, $P < 0.0001$). To assess cell growth and aromatic consumption, the variations in the changes of cell density (OD) and aromatic concentration (Con) of the mutants were compared against those of the WT control (★, $P < 0.05$ for OD; ☆, $P < 0.05$ for Con; two-tailed mixed model ANOVA with Sidak's multiple comparisons). All values represent the mean of triplicate cultures, with error bars depicting the standard deviation from that mean.

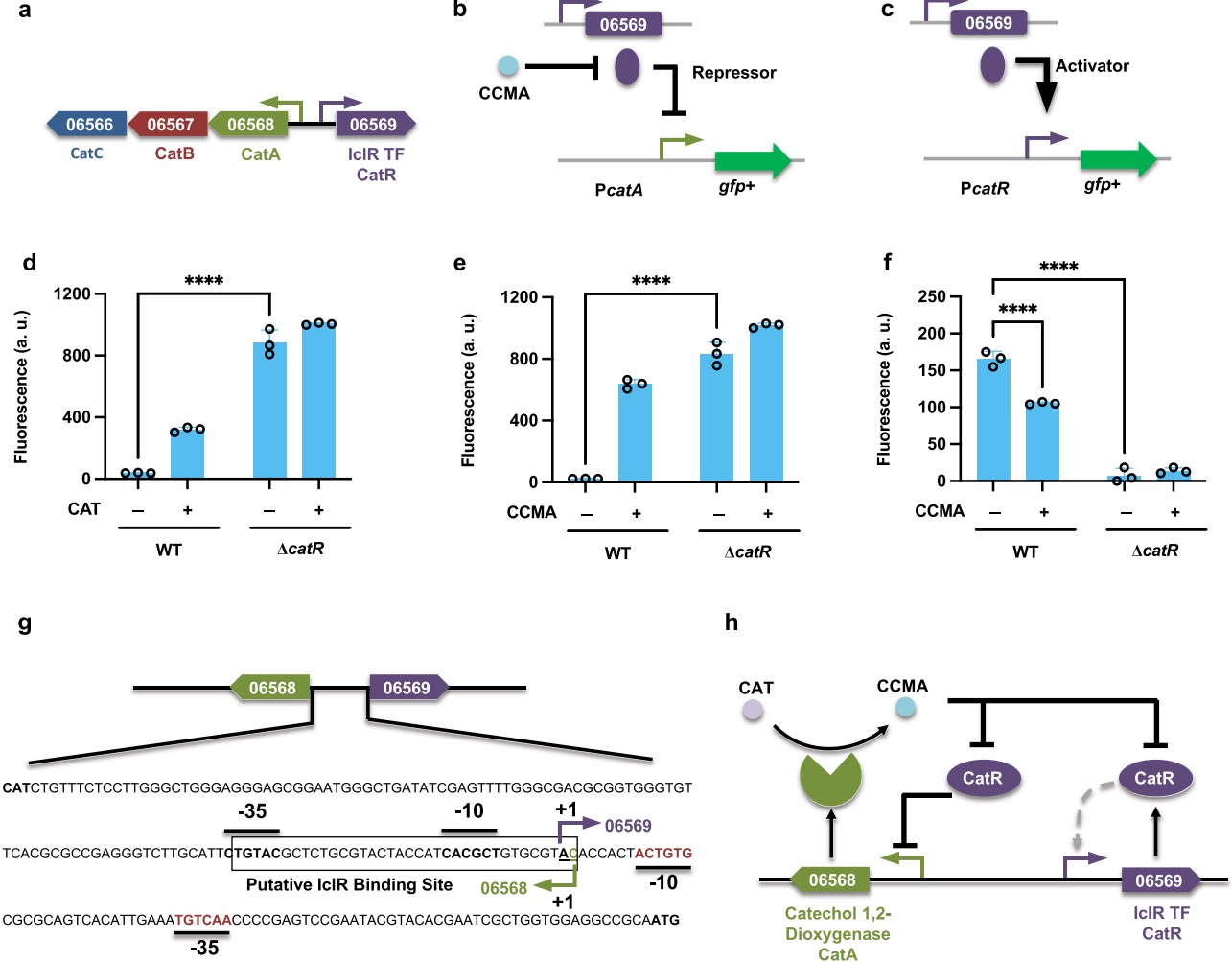

**Fig. 5 Functional analysis of the transcription factor *catR* (LPD06569).** **a** A catechol (CAT) degradation operon with an annotated transcriptional factor (TF). Promoters are represented as small arrows. Genes are shown with LPD gene numbers from the NCBI database (Refseq, CP003949.1). For the full annotations of the pathway genes, see Supplementary Table 1. **b**, **c** The proposed models for the role of *catR*. **d**, **e** To analyze the function of *catR* in regulating the CAT degradation cluster, the fusion construct P*catA*-GFP+ was expressed in both the WT and the TF deletion mutant strains. Normalized fluorescence of the WT and Δ*catR* strains is shown in response to CAT and *cis-cis* muconate (CCMA). **f** The fusion construct P*catR*-GFP+ was used to analyze the self-regulation pattern of *catR*. Normalized fluorescence of the WT and Δ*catR* strains is shown in response to CCMA. All cultures contained 1 g/L glucose with (+) or without (−) the respective aromatic carbon source (CAT 0.3 g/L and CCMA 5 g/L). All the fluorescence values were determined in the early stationary phase and normalized to $OD_{600}$. Unpaired two-tailed *t* test was used to compare the variations in the changes of fluorescence of the strains under different conditions (****, $P < 0.0001$). All values represent the mean of triplicate cultures, with error bars depicting the standard deviation from that mean. **g** The potential regulatory sequences from the intergenic region between *catR* and *catA*. All the regulatory features and sequence elements were identified by comparison with the position identified in *Rhodococcus erythropolis*[37]. Transcriptional start sites (TSS, +1) are underlined, and transcriptional initiation is indicated by bent arrows. The proposed −35 and −10 regions are in bold, and the putative IclR-type regulator binding site is boxed. **h** The deduced schematic of the regulation model of the CAT degradation pathway in *R. opacus*.

induction prevention. This finding indicates that diverse CCR mechanisms have evolved in *R. opacus* to deal with mixtures of several substrates, an idea that needs to be further investigated.

## Discussion

Unraveling the complex transcriptional regulation of the catabolism of aromatics in *R. opacus* is a prerequisite for engineering this promising chassis for many biotechnological applications. It was discovered that many related catabolic pathways did not carry the same regulatory system, suggesting that the regulatory systems and their target operons seem to become associated independently, which makes the regulatory system varied and complex[26]. In this study, the AraC-type regulators we found for controlling the expression of the phenol and the guaiacol

funneling pathways are transcriptional activators (Figs. 1 and 2). Our results also demonstrated that phenol is the effector compound for inducing expression of the *pheR2-pheB2A2* cluster, but intriguingly, there is evidence to suggest that *cis-cis* muconate is also a ligand for TF *pheR2* (Supplementary Fig. 1b). While unexpected, this behavior is not unprecedented: our multi-omics data have revealed that the corresponding phenol degradation cluster (*pheB2A2*) is also highly upregulated when treated with guaiacol (121-477-fold)[18], another compound which is degraded through the CAT branch of the β-ketoadipate pathway.

IclR-type TFs are generally recognized as transcriptional repressors[26]; however, assays of promoter activity in both the WT and the mutant strains indicated that the IclR-type TF *hbaR1* and *vanR1* both work as transcriptional activators in regulating the expression of 4-hydroxybenzoate and vanillate

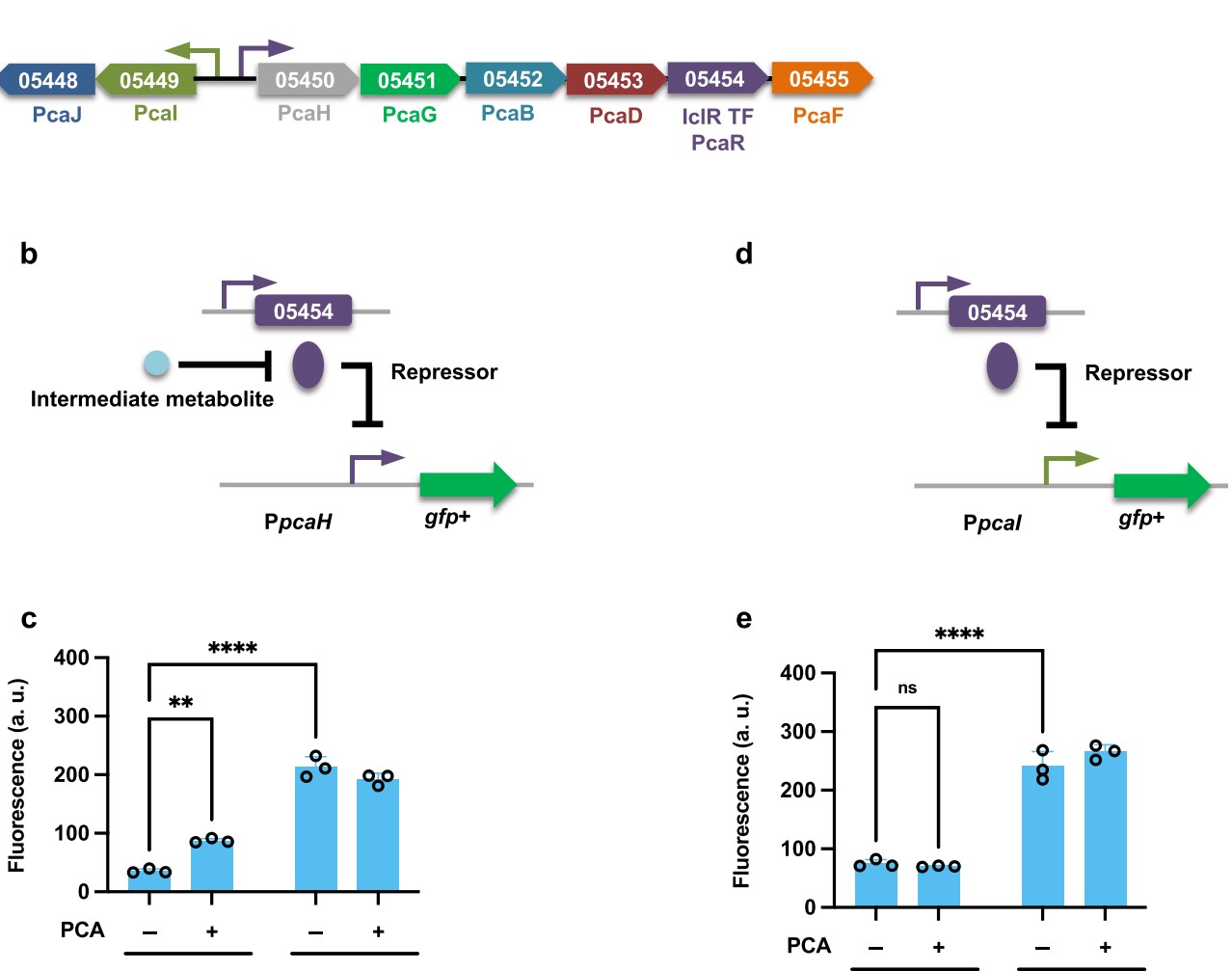

**Fig. 6 Transcriptional regulation of the protocatechuic acid branch of the β-ketoadipate pathway. a** A protocatechuic acid (PCA) degradation operon with an annotated transcription factor (TF). Promoters are represented as small arrows. Genes are shown with LPD gene numbers from the NCBI database (Refseq, CP003949.1). For the full annotations of the pathway genes, see Supplementary Table 1. **b**, **d** The proposed models for the role of *pcaR* (LPD05454). **c**, **e** To analyze the function of *pcaR*, the fusion construct P*pcaH*-GFP+ and P*pcaI*-GFP+ were expressed in both the WT and the corresponding TF deletion mutant strains. The normalized fluorescence of the WT and Δ*pcaR* strains was measured with and without supplemental PCA. All cultures contained 1 g/L glucose with (+) or without (−) 0.3 g/L PCA. All the fluorescence values were determined in the early stationary phase and normalized to OD$_{600}$. Unpaired two-tailed *t* test was used to compare the variations in the changes of fluorescence of the strains under different conditions (**, $P < 0.0021$; ****, $P < 0.0001$; ns, not significant). All values represent the mean of triplicate cultures, with error bars depicting the standard deviation from that mean.

funneling pathways, respectively (Figs. 3 and 4). This is not inconsistent with previous data, as IclR-type TFs have also been found to work as activators in regulating the catabolic pathways[42]. In addition, our findings also demonstrated that the transcriptional regulation of the two different branches of the β-ketoadipate pathway is controlled by separate IclR-type transcriptional repressors (Figs. 5 and 6). For the CAT branch, two regulatory modes have been reported: when CAT-degradation genes are controlled by LysR-type activators, the effector compound is usually the intermediate *cis-cis* muconate, whereas in operons under the regulation of IclR-type repressors, this role is mostly fulfilled by aromatic substrates[37]. Although the IclR-type transcriptional regulators have a similar structure as the LysR-type regulators[43], the rather dissimilar amino acid sequences distinguish these two families. In this study, our results

confirmed that the effector compound for induction of the CAT degradation pathway is *cis-cis* muconate, which is dramatically different from that of closely-related strain *R. erythropolis* CCM2595, where the expression of CAT-degradation genes is induced by phenol, rather than CAT or *cis-cis* muconate[37]. Although, the entire PCA branch of β-ketoadipate pathway consists of 8 individual genes, our analysis indicated a single, IclR-type repressor, *pcaR* (LPD05454), is responsible for regulating the entire PCA operon (Fig. 6). This is also a departure from precedent, as for example, the mycobacterium *C. glutamicum* places the PCA degradation pathway under the control of two different regulators: *pcaIJ* and *pcaFDO* are regulated by the IclR-type repressor *pcaR*, whereas the expression of *pcaHG* is controlled by an atypically large ATP-binding LuxR family (LAL)-type activator *pcaO*[38,44].

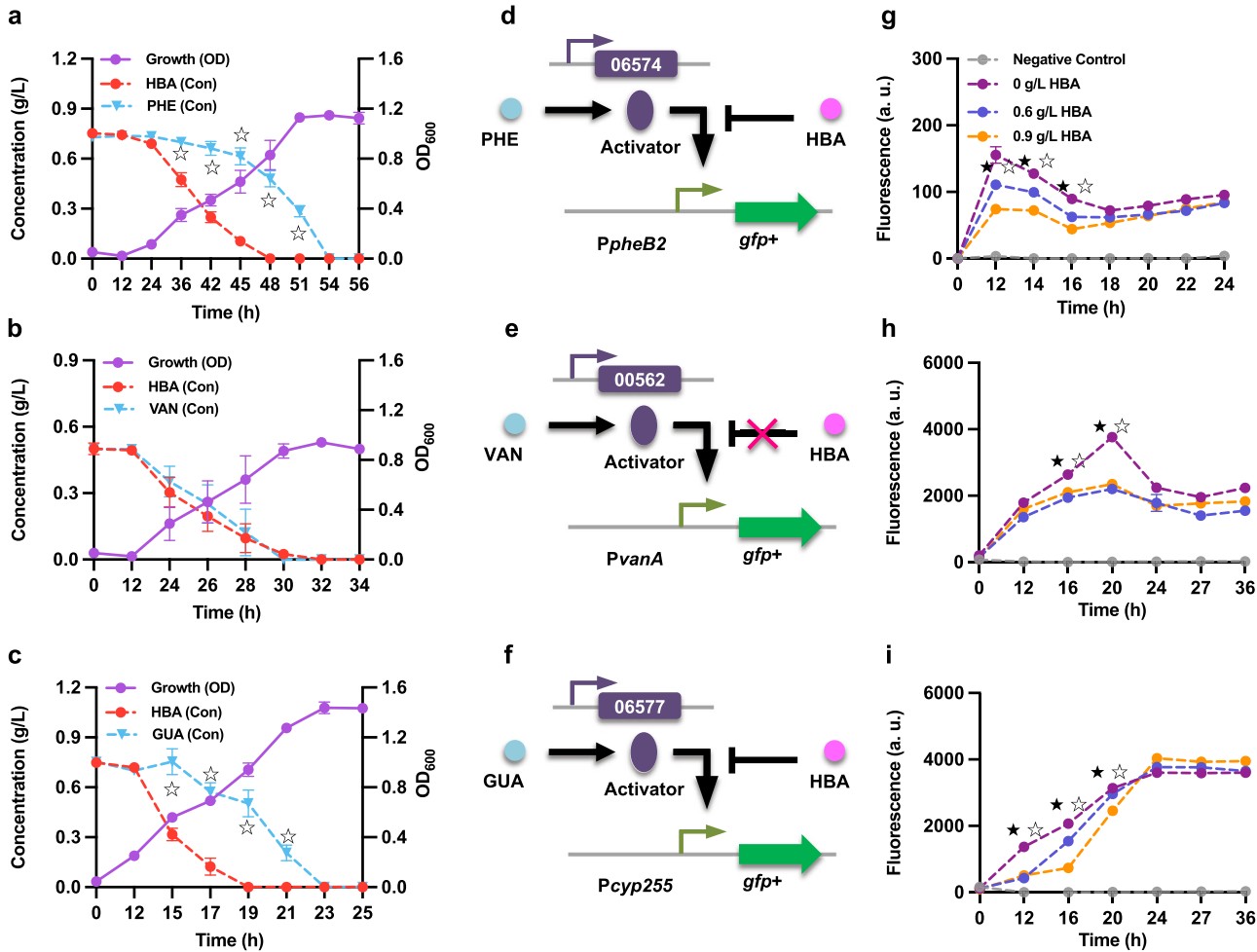

**Fig. 7 Preferential utilization of 4-hydroxybenzoate (HBA) over phenol (PHE) and guaiacol (GUA) by _R. opacus_. a–c** Utilization profiles of the aromatic mixtures. A glucose-grown WT cell culture was used as the inoculum. Cells were cultured in minimal medium with different aromatic mixtures as carbon sources: **a** 0.75 g/L HBA + 0.75 g/L PHE; **b** 0.5 g/L HBA + 0.5 g/L VAN; **c** 0.75 g/L HBA + 0.75 g/L GUA. At each time point, a 200 μL cell suspension was taken to measure the cell density (OD), and concentrations of the aromatics (Con) in the supernatant were determined by HPLC. **d–f** The proposed working models for the interference between HBA and PHE (**d**), HBA and VAN (**e**), and HBA and GUA (**f**). The different transcriptional constructs were transformed into WT _R. opacus_ strain and used for fluorescence assays. **g–i** The normalized fluorescence in response to the aromatic mixtures was measured for each mixture: **g** HBA + PHE, **h** HBA + VAN, and **i** HBA + GUA. All cultures (**g–i**) contained 1 g/L glucose, with either 0.3 g/L PHE, 0.3 g/L VAN, or 0.3 g/L GUA. The concentrations of HBA were set at 0, 0.6, and 0.9 g/L. All of the fluorescence values were determined and normalized to $OD_{600}$. Two-tailed mixed model ANOVA with Sidak's multiple comparisons was used to compare the variations in the changes of concentration (Con) of the two aromatic compounds (☆, $P < 0.05$); for fluorescence assays, the variations in the changes of fluorescence of the two-compound treatment were compared against that of the one-compound treatment only (★, $P < 0.05$ for 0.6 g/L HBA; ☆, $P < 0.05$ for 0.9 g/L HBA; two-tailed mixed model ANOVA with Sidak's multiple comparisons). All values represent the mean of triplicate cultures, with error bars depicting the standard deviation from that mean.

In general, the coding gene for an IclR-type TF lies upstream of its target gene cluster and is transcribed in the opposite direction[26]. In this study, however, we found that the location and transcription orientation of IclR-type regulators varied. For instance, the _catR_ regulator is located upstream of the CAT degradation operon, and is transcribed divergently (Fig. 5a); however, the upstream-located _hbaR1_ is transcribed in the same orientation as the critical gene in the 4-hydroxybenzoate funneling pathway (Fig. 3a). Notably, the _vanR1_ and _pcaR_ are located within their target operons (Figs. 4a and 6a), arranged in their genomic neighborhoods in a way which may provide extra regulatory functions. _vanR1_ is located immediately downstream of the vanillate monooxygenase reductase _vanB_ (LPD00563) and shares a promoter with this operon (Fig. 4a), an arrangement which suggests that the vanillate degradation process may be regulated through a positive feedback loop (PFL)[45]. While studies have demonstrated that the positive-feedback response to an

environmental signal is slower than in those systems that produce the regulatory protein constitutively[46], this moderate delay could be beneficial in _R. opacus_ as a means of ordering the action of cellular response mechanisms in time, e.g., by upregulating the gene clusters involved in one-carbon compound metabolism to prepare for the harmful formaldehyde released during a deme-thylation step of vanillate catabolism. For _pcaR_, however, the regulator lies in the middle of the _pcaH-pcaF_ (LPD05450-05455) coding region and thus shares the promoter, resulting in a negative feedback loop (NFL). Because some of the _pca_ genes (_e.g._, _pcaIJ_ and _pcaFD_) are also involved in the degradation of CAT, this NFL allows the enhanced PcaR to downregulate the expression of those genes, which consequently serves as a meta-bolic node for controlling the carbon flux of the β-ketoadipate pathway towards the TCA cycle to reduce succinate overflow[47].

In nature, carbon and energy resources are often limited. Thus, specific bacteria that are more efficient or more selective in

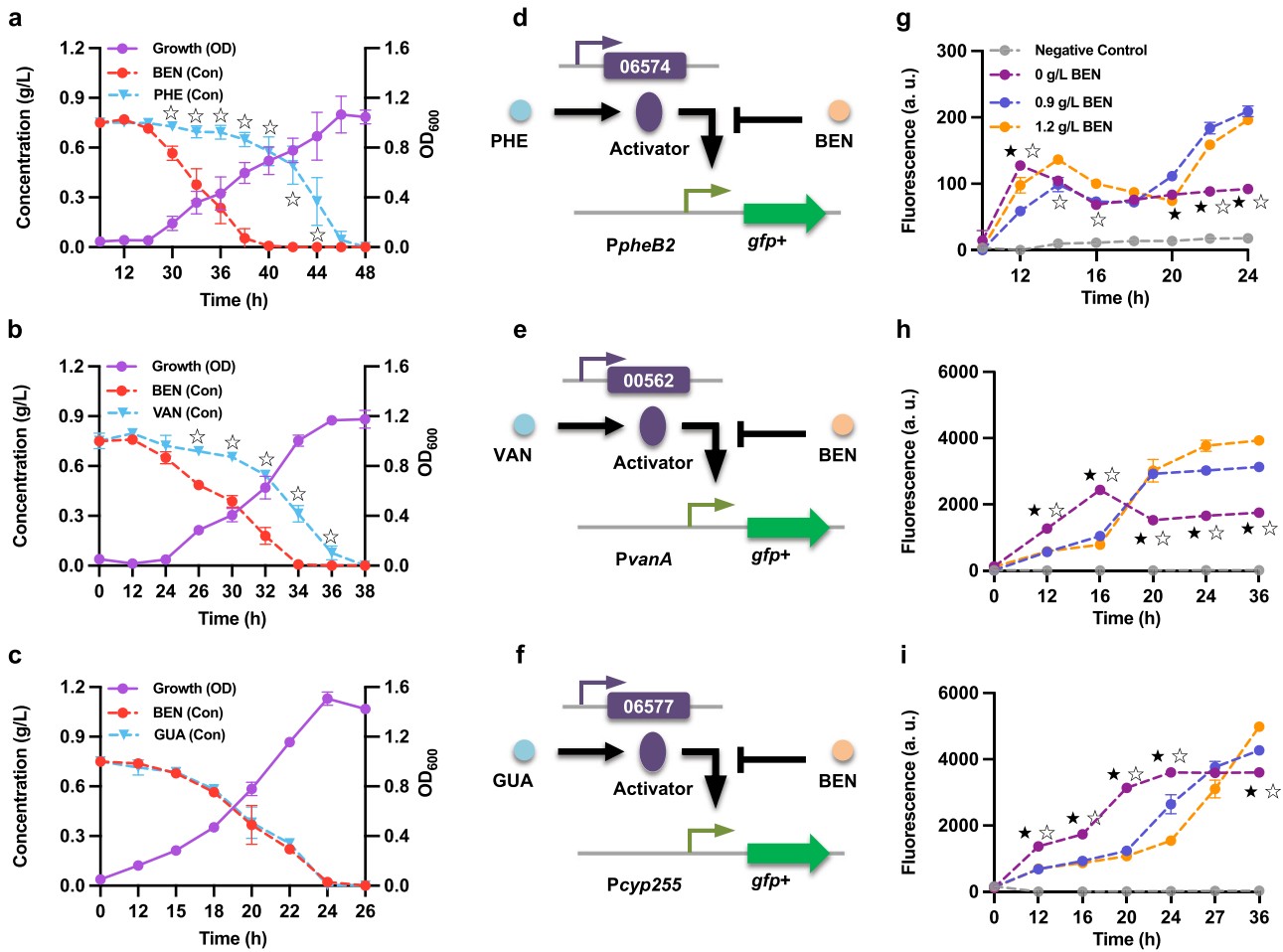

**Fig. 8 Preferential utilization of benzoate (BEN) over phenol (PHE) and vanillate (VAN) by *R. opacus*. a–c** Utilization profiles of the aromatic mixtures. A glucose-grown WT cell culture was used as the inoculum. Cells were cultured in minimal medium with different aromatic mixtures as carbon sources: **a** 0.75 g/L BEN + 0.75 g/L PHE, **b** 0.75 g/L BEN + 0.75 g/L VAN, and **c** 0.75 g/L BEN + 0.75 g/L guaiacol (GUA). At each time point, a 200 μL cell suspension was taken to measure the cell density (OD), and concentrations of the aromatics (Con) in the supernatant were determined by HPLC. **d–f** The proposed working models for the interference between BEN and PHE (**d**), BEN and VAN (**e**), and BEN and GUA (**f**). The different transcriptional constructs were transformed into the WT *R. opacus* strain and used for the fluorescence assays. **g–i** Normalized fluorescence was measured in response to the aromatic mixtures: **g** BEN + PHE, **h** BEN + VAN, and **i** BEN + GUA. All cultures (**g–i**) contained 1 g/L glucose with either 0.3 g/L PHE, 0.3 g/L VAN, or 0.3 g/L GUA. The concentrations of BEN were set at 0, 0.9, and 1.2 g/L. All of the fluorescence values were determined and normalized to $OD_{600}$. Two-tailed mixed model ANOVA with Sidak's multiple comparisons was used to compare the variations in the changes of concentration (Con) of the two aromatic compounds (☆, $P < 0.05$); for fluorescence assays, the variations in the changes of fluorescence of the two-compound treatment were compared against that of the one-compound treatment (★, $P < 0.05$ for 0.9 g/L BEN; ☆, $P < 0.05$ for 1.2 g/L BEN; two-tailed mixed model ANOVA with Sidak's multiple comparisons). All values represent the mean of triplicate cultures, with error bars depicting the standard deviation from that mean.

utilizing the carbon sources in their habitat will have significantly higher growth rates and therefore greater competitive success than other microorganisms[48]. In this work, we found that benzoate is the most-preferred substrate among the model compounds we tested. Additionally, we were able to firmly establish the substrate hierarchy of all tested compounds in *R. opacus* (Figs. 7–8 and Supplementary Figs. 5–9). The preference for benzoate can presumably be traced to the different energetic demands of the funneling pathways—the conversion of benzoate to CAT consumes no net reducing equivalents because the NADH oxidized in the first step is recovered during the next dehydrogenation reaction by an NAD[+]-dependent dehydrogenase. In contrast, the conversions of 4-hydroxybenzoate or vanillate to PCA and phenol or guaiacol to CAT require the oxidation of NAD(P)H[18,34]. Energetic considerations cannot, however, explain the preference for 4-hydroxybenzoate, since the initial steps in the metabolisms of 4-hydroxybenzoate, phenol, and guaiacol have similar requirements for reducing equivalents. To facilitate

understanding the complex intracellular dynamics, a cybernetic model has been previously developed to describe the resource allocation and microbial kinetics that influence the hierarchical utilization of carbon sources[49]; different variants of this model have been able to account for a variety of instances of preferential carbon uptake in *E. coli*[50]. From this resource-allocation point of view, it is reasonable to find that *R. opacus* prefers 4-hydroxybenzoate over phenol and guaiacol. Compared to phenol, which has redundant funneling pathways, and guaiacol, which requires an accessory demethylation pathway to detoxify formaldehyde released from the funneling process, the 4-hydroxybenzoate funneling pathway's "single regulator-single enzyme" unit is relatively simple and thus less resource-intensive.

Intriguingly, although the vanillate and guaiacol funneling pathways both require reducing equivalents and the accessory demethylation pathway, our results indicated preferential utilization of vanillate over guaiacol (Supplementary Fig. 9). This preference cannot be explained by the two theories discussed

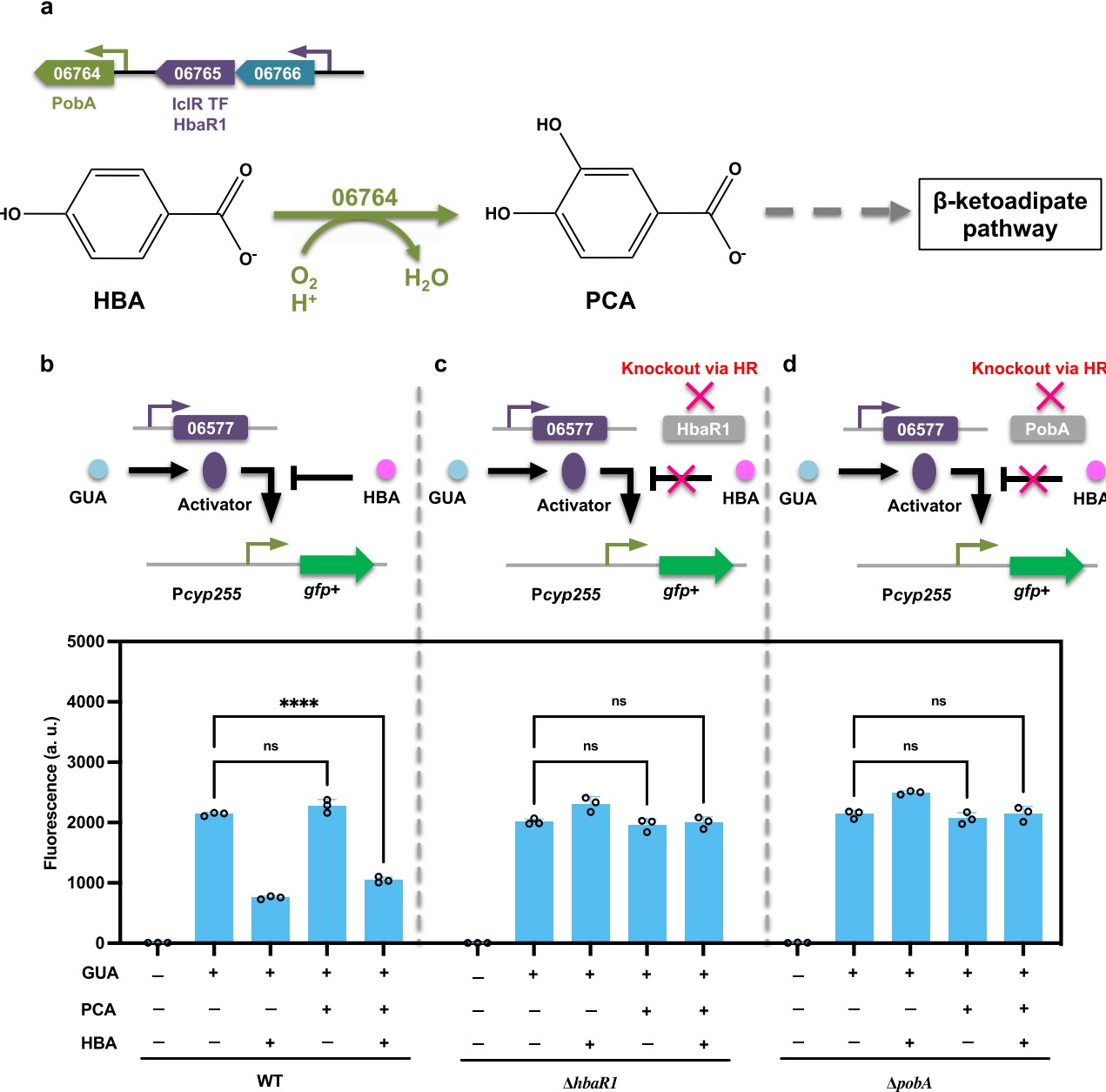

**Fig. 9 Promoter activity of P*cyp255* in response to guaiacol and 4-hydroxybenzoate. a** Schematics depicting the funneling pathway for 4-hydroxybenzoate (HBA) and the annotated transcription factor. **b–d** Proposed working models for the promoter P*cyp255* during growth in minimal medium with guaiacol (GUA) and HBA (upper) and fluorescence of P*cyp255*-GFP+ in response to GUA and HBA in WT and knockout strain backgrounds (lower). All cultures contained 1 g/L glucose with either 0.3 g/L GUA, or 0.3 g/L GUA plus 0.9 g/L protocatechuic acid (PCA) or 0.9 g/L HBA. The concentrations of the PCA and HBA mixtures were set at 0.45 g/L. All the fluorescence values were determined in the early stationary phase and normalized to OD$_{600}$. Unpaired two-tailed $t$ test was used to compare variations in the changes of fluorescence of the two aromatic mixtures against that of the GUA-only control (****, $P < 0.0001$; ns, not significant). All values represent the mean of triplicate cultures, with error bars depicting the standard deviation from that mean.

above, but the difference in the structures of the two funneling pathway operons may indicate that the preference is attributable to the mechanistic model[51]. Specifically, in a mixture of vanillate and guaiacol, the proposed PFL regulation pattern for vanillate keeps the TF *vanR1* preferentially expressed, which in turn could further promote the induction of the vanillate funneling pathway, supporting fast cell growth. Fast growth on vanillate may result in dilution of *guaR*, thus preventing effective induction of the guaiacol funneling pathway. While these models developed in *E. coli* can be used to provide a coarse-grained description of the preferential utilization of aromatics, we believe that a new

generation of models, specifically and precisely tuned for *R. opacus*, is still needed.

Our results also provide some insights into the mechanism of the substrate hierarchy in *R. opacus*, or more specifically, the transcriptional cross-regulation of the funneling pathways. Induction prevention has been used to describe the preferential utilization of aromatics in other strains[31–33]. However, our results showed that in the mutants Δ*hbaR1* and Δ*pobA* (for which degradation of 4-hydroxybenzoate was abolished), no transcriptional repression of the phenol or guaiacol funneling pathway was observed (Fig. 9 and Supplementary Fig. 10), suggesting that the

preference for 4-hydroxybenzoate cannot be explained by induction prevention alone. Inspired by the resource allocation view, we speculate that this preference might be controlled by a novel global regulation mechanism. More specifically, due to the relative simplicity of the 4-hydroxybenzoate funneling pathway, the investment towards the utilization of 4-hydroxybenzoate is much smaller, which allows cells to maximize profit. In this scenario, once cells detect 4-hydroxybenzoate in the environment, more resources are allocated to the 4-hydroxybenzoate degradation pathway, with transcription and translation of the operons which enable the degradation of phenol or guaiacol consequently decreased, resulting in the sequential utilization of the three compounds. Transporter-mediated inducer exclusion has been used to describe the molecular mechanism of the substrate hierarchy in *Bacillus* species[29]. In this study, we also studied the transcriptional regulation of the annotated shikimate transporter LPD06699 and found that the expression of this transporter could be induced by all the model compounds we tested; furthermore, this upregulation was enhanced in the TF deletion mutant (ΔLPD06698) (Supplementary Figs. 11–12). Interestingly, in the vanillate scenario alone, the upregulation of this transporter improved carbon utilization (Supplementary Fig. 12e), supporting the hypothesis that inducer exclusion might be one of the mechanisms responsible for the substrate hierarchy in *R. opacus* that still needs to be evaluated.

In conclusion, we identified and investigated the TFs involved in regulating several aromatic degradation pathways in *R. opacus* by combining gene knockouts with aromatic sensors. We also observed that individual lignin model compounds in an aromatic mixture are consumed by *R. opacus* in sequential order and that this preferential utilization pattern can be ascribed to the transcriptional cross-regulation of the funneling pathways. While we have been able to describe many mechanisms of individual pathway control, aspects of how these pathways interact have yet to be fully explained. Nonetheless, our results can inform the development of strain-specific models of *R. opacus* metabolism for industrial applications.

## Methods

**Strains and plasmids**. *R. opacus* PD630 (DSMZ 44193) was used as the strain (WT) to construct mutants. Unless otherwise indicated, cells were grown at 30 °C and 250 rpm in a previously described minimal salts medium B[52] with different carbon sources. All the plasmids constructed in this study were confirmed by DNA sequencing and are summarized in Supplementary Table 2.

**Plasmid construction**. For constructing the TF deletion strains, the deletion construct was assembled as described previously[53]. ~600 bp regions up- and downstream of the selected transcriptional regulator were amplified from the genome. These two pieces of DNA fragments were assembled into a plasmid containing an *E. coli* origin of replication (p15a) and a chloramphenicol selection cassette. To analyze the potential function of the selected TF, the upstream region containing the promoter and ribosome-binding site (RBS) of the proposed target genes was cloned in front of *gfp+*. The resulting transcriptional construct was then expressed in both WT and the corresponding TF deletion mutant strains. Unless otherwise described, all plasmids were assembled in *E. coli* DH10B using Gibson Assembly[54].

**T7 RNAP-based CRISPRi**. The T7 RNAP expression platform was developed in our previous study[36]. Specifically, the T7 RNAP gene was integrated into the *R. opacus* chromosomal neutral site (ROCI3) under the control of the phenol inducible promoter (PpheB2). The codon-optimized *dcas9* from *Streptococcus thermophilus* (*dcas9Sth1*) was placed under the control of the pBAD promoter. The expression of the guide RNA was driven by T7 promoter. One sgRNA (sgRNA PHE_1) was designed to target *pheB2* (LPD06575), and a strain with *dcas9Sth1* and T7 RNAP but no sgRNA was used as a control. The uninduced condition represented 0 mM arabinose and 0.3 g/L PHE, while the induced condition represented 50 mM arabinose and 0.3 g/L PHE.

**Transformation of *R. opacus***. Competent cells were made as described previously[52]. Cells were transformed with ~500 ng plasmid DNA by electroporation. To generate *R. opacus* knockout mutants, a previously developed homologous recombination method was applied with modifications[53]. Briefly, a helper plasmid expressing modified viral recombinases was introduced into the strain via electroporation, and electrocompetent cells were made from this strain. The cells were transformed with ~2 μg suicide plasmid DNA. Positive colonies were verified by colony PCR. All the strains used in this study are summarized in Supplementary Table 3.

**Cell growth and fluorescence measurements**. The optical density at 600 nm ($OD_{600}$), absorbance at 600 nm ($Abs_{600}$), and fluorescence were measured using a Tecan Infinite M200 Pro plate reader. For the measurements of GFP+ fluorescence, the excitation and emission wavelengths were 488 and 530 nm, respectively[52]. Fluorescence values were normalized using Eq. (1):

$$\text{Fluorescence}_{norm} = \frac{\text{Fluorescence}_{sample}}{A600_{sample}} - \frac{\text{Fluorescence}_{control}}{A600_{control}} \qquad (1)$$

where $\text{Fluorescence}_{norm}$ is the normalized fluorescence, $A600_{sample}$ is the absorbance (at 600 nm) of the test strain, $\text{Fluorescence}_{sample}$ is the fluorescence value of the test strain, $A600_{control}$ is the absorbance of the empty vector control strain, and $\text{Fluorescence}_{control}$ is the fluorescence value of the empty vector control strain.

**Aromatic consumption profiling**. To measure the aromatics, 200 μL samples from each cell culture were centrifuged at 3500 rcf for 5 min, and the cell culture supernatant was subsequently analyzed. For cell cultures using a single aromatic compound as the carbon source, the concentration of each lignin model compound was measured by comparing UV absorbance values to a standard curve for each lignin model compound. For cultures using the aromatic mixture, overnight cultures of WT *R. opacus* cells grown on 1 g/L glucose were harvested, washed, and then resuspended in fresh minimal medium containing different combinations of aromatic compounds at an initial $OD_{600}$ of ~0.05. Aromatics in the culture supernatant were detected using an Agilent 1260 Infinity II HPLC system equipped with the Agilent Poroshell 120 EC-C18 column (4.6 × 100 mm, 2.7 μm) and a UV detector (at 280 nm). The temperature was set at 60 °C, and the flow rate was 1 mL/min. Mobile phase A (water with 0.1% formic acid) and mobile phase B (acetonitrile with 0.1% formic acid) were used as follows (A%/B% with a gradient elution): 92/8 at 0 min, 74/26 at 5 min, 50/50 at 8 min, and 92/8 at 10 min. Concentrations were determined by comparing UV absorbance values to a standard curve for each lignin model compound.

**Statistics and reproducibility**. All experiments were conducted with at least three biological replicates. Differences between the control and engineered strains were analyzed using GraphPad Prism 9.

**Reporting summary**. Further information on research design is available in the Nature Research Reporting Summary linked to this article.

## Data availability

The main data and raw data supporting the results described in this study are available within the main paper, its Supplementary Information, Supplementary Data 1, and Supplementary Data 2. Additional data generated and/or analyzed during the current study are available from the corresponding author upon reasonable request.

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

## Acknowledgements

The authors thank Dr. Shulin Chen for pAL5000 (S) and KanR, Dr. Anthony Sinskey for GentR, Dr. Graham Hatfull for the Che9c recombinases, and Dr. Sarah Fortune for the dCas9Sth1 gene. The authors also thank James Ballard for his comments on the manuscript. This work was supported by the United States Department of Energy (DE-SC0018324 to T.S.M.) and the United States Department of Agriculture (2020-33522-32319 to T.S.M.).

## Author contributions

T.S.M. conceived the project. J.D. and T.S.M. designed experiments and analyzed the data. J.D. performed the experiments. J.D., R.C., and T.S.M. wrote the manuscript.

## Competing interests

The authors declare no competing interests.
