## [Peer Review File · Communications Biology]

Reviewers' comments:

Reviewer #1 (Remarks to the Author):

Diao et al. study transcriptional regulators involved in phenol, guaiacol and benzoate degradation in *Rhodococcus opacus*. The authors use knock-out strains and GFP-based reporter assays to identify the regulator genes for the respective pathways. The designate several transcription factors to their respective target genes/operons and degradation pathways. However, the style of writing is very hard to follow (e.g. using the gene numbers to designate the mutants instead of reasonable gene names), and the results part contains too much (and repetitive) methods sections to be understood properly. After spending a lot of time with this review, I send it back half-finished because I just had to give up understanding what the authors may have done in their experiments and may have meant in their interpretations.

Individual comments:

The title of this study is completely misleading and has nothing to do with what is investigated.

Line

60ff the significance statement does not contain any new aspect and is simply a repetition of the abstract

138 the cited references do not contain studies on aromatic compounds. Moreover, there are plenty of studies available on the regulation of aromatic degradation, including catabolite repression (aerobic and anaerobic), so they are not "rare", although certainly less frequent than on sugars.

172 it makes no sense that these studies should help in "conversion of lignocellulose ..."

190 PHE does not need to be abbreviated, this is only misleading for the reader; same for CCMA, GUA and BEN.

191 ff the authors use the gene number from the genome sequence to designate the genes of interest and the mutants produced (such as LPD06740 etc). This makes the paper almost unreadable and may be regarded as insult to the reader and particularly the reviewer, and therefore must be changed to some more memory-friendly nomenclature to make the paper acceptable.

208 it needs to be better specified what reporter was tested with what mutant

225 the description of the CRISPRi experiment contains too much technical detail (that should be mentioned in Methods), but does not clearly state the result of the experiment. After reading over the passage several times, I still don't get the message.

235 the potential crosstalk is also described in way that is hardly comprehensible.

238 a promotor (if that is meant with pLPD06740 ?) is not a "PHE specialist" or PHE sensor!

245 "TF knockout" is lab jargon

264 give a number for the decrease and point out the significance level, The effect of the mutant is so small that this is certainly not the main /direct regulator involved. Moreover, the results should not be hidden in the supplement

270 identities of 26% mean that the proteins are totally different from each other. Therefore, it is clear that these mutants also had no effect. The mentioned "significant decrease" is very questionable, since the about same reporter activity in all mutants indicates that the decrease is caused by some technical difference between the WT and mutant experiments (presence/absence of antibiotics or anything else???).

276 what is meant with "significant induction is found" in the triple mutant??? I see (almost) no difference between WT and any of the mutants.

286 the lengthy description of how the reporter was made belongs into Methods (moreover, it is the same procedure as before once again)

295 why all the jumping between the figures (supp. Fig 1e here, but supp. Fig. 3 in the previous chapter)

326 ff I am totally giving to follow what the authors did with the "funneling pathways". The way the experiments and results are described, I can not figure out what the authors did and how to judge the results. The paper needs to be completely re-written with reasonable gene designations and

separation of methods and results.

Discussion: much too long (& pages) and mostly repeats what has been stated in Results.

Methods: almost nothing is in there. A lot of superfluous stuff from the Results should be described here.

Reviewer #2 (Remarks to the Author):

The authors have used gene knockout strategies to identify and assign activator and repressor roles to transcription factors that enable a better understanding of transcriptional regulation in *Rhodococcus opacus* relevant to aromatic degradation pathways for lignin valorisation. The work is described well and can be seen as a valuable addition to the efforts of developing synthetic biology strategies for lignocellulosic valorisation efforts and beyond. Whilst the work appears to be sufficiently novel to merit publication, the generic title of the manuscript is a bit misleading. I would urge the authors to think of a more meaningful title that specifically reflects the work carried out, including the name of the organism they have worked with. It is advisable to have a title that reflects the key specific findings of the research. It would also help for the authors to state clearly the specific findings and how this advances the field. The summarised findings in the abstract come across as being a bit too general, lacking in specifics. Perhaps, it would help if the authors could be more specific about the novel findings of the investigation in the abstract and the title. Otherwise, I believe the authors have done a good job.

Reviewers' comments:

Reviewer #1 (Remarks to the Author):

Diao et al. study transcriptional regulators involved in phenol, guaiacol and benzoate degradation in Rhodococcus opacus. The authors use knock-out strains and GFP-based reporter assays to identify the regulator genes for the respective pathways. The designate several transcription factors to their respective target genes/operons and degradation pathways. However, the style of writing is very hard to follow (e.g. using the gene numbers to designate the mutants instead of reasonable gene names), and the results part contains too much (and repetitive) methods sections to be understood properly. After spending a lot of time with this review, I send it back half-finished because I just had to give up understanding what the authors may have done in their experiments and may have meant in their interpretations.

Individual comments:

The title of this study is completely misleading and has nothing to do with what is investigated.

R: Thanks for your comment. We have changed the title, and more specific details have been included to make it more accurate to reflect what we investigated in this study.

Line

60ff the significance statement does not contain any new aspect and is simply a repetition of the abstract

R: Thanks for your comment. As a significance statement is not required for *Communications Biology*, we just removed this part from our revised manuscript.

138 the cited references do not contain studies on aromatic compounds. Moreover, there are plenty of studies available on the regulation of aromatic degradation, including catabolite repression (aerobic and anaerobic), so they are not “rare”, although certainly less frequent than on sugars.

R: Thanks for your suggestions. These references (28-30) are cited for the studies demonstrating the CCR between sugar mixtures and non-sugar substrates. We are sorry for putting them in the inappropriate place. We have relocated these citations in our revised manuscript. As we have discussed, we agree with you that they are not “rare”. Therefore, we rephrased this sentence in our revised manuscript.

172 it makes no sense that these studies should help in “conversion of lignocellulose ...”

R: Thanks for your comment. Lignocellulose consists of carbohydrate polymers (e.g., cellulose and hemicellulose) and aromatic polymers (e.g., lignin), and recent research indicates that converting lignin to high-value fuels and chemicals would significantly improve the overall competitiveness of biorefineries. However, due to the structural heterogeneity of lignin, the depolymerization process typically results in diverse aromatic products, which are challenging to valorize. The results we demonstrate here will advance our understanding of the mechanisms underlying the complex regulation of aromatic catabolism, and facilitate the development of *R. opacus* as a chassis for valorizing lignin, which will make the conversion of lignocellulose more economically profitable.

190 PHE does not need to be abbreviated, this is only misleading for the reader; same for CCMA, GUA and BEN.

R: Thanks for your suggestions. We changed to use the full name to describe the aromatic compounds throughout our revised manuscript.

191 ff the authors use the gene number from the genome sequence to designate the genes of interest and the mutants produced (such as LPD06740 etc). This makes the paper almost unreadable and may be regarded as insult to the reader and particularly the reviewer, and therefore must be changed to some more memory-friendly nomenclature to make the paper acceptable.

R: Thanks for your comments and suggestions. We agree with you that using the gene number to designate the transcription factors is not easy to understand. In the revised manuscript, we used a more memory-friendly nomenclature to designate all the transcription factors. We also provided the detailed annotations of the pathway-specific genes in Supplementary Table 1, and hope these efforts could make our paper more acceptable.

208 it needs to be better specified what reporter was tested with what mutant

R: Thanks for your suggestion. We have rephrased those sentences and specified which promoter was tested in which TF deletion mutant, making this passage more understandable.

225 the description of the CRISPRi experiment contains too much technical detail (that should be mentioned in Methods), but does not clearly state the result of the experiment. After reading over the passage several times, I still don't get the message.

R: Thanks for your suggestion. We have moved the technical details of the CRISPRi experiment into the Methods section. We also revised this passage substantially with the changed nomenclature, and hope these revisions could make it more understandable.

235 the potential crosstalk is also described in way that is hardly comprehensible.

R: Thanks for your comment. Our result showed that in $\Delta pheR1$, expression from promoter *PpheB1* was severely inhibited but still detectably ON (Fig. 1d). We also found that in a dual mutant ($\Delta pheR1 \Delta pheR2$), the fluorescence output of the promoter *PpheB1* was null, suggesting that this promoter may have crosstalk with TF *pheR2*. To make the discussion of the crosstalk more acceptable, we moved this part to the end of the second paragraph of the Results section, following the demonstration of promoter activities (*PpheB1* and *PpheB2*) in the two different TF deletion mutants ($\Delta pheR1$ and $\Delta pheR2$).

238 a promotor (if that is meant with pLPD06740?) is not a "PHE specialist" or PHE sensor!

R: Thanks for your comment. We have removed this sentence from our revised manuscript.

245 "TF knockout" is lab jargon

R: Thanks for your comment. We have changed "TF knockout" to "TF deletion mutant" to address this comment.

264 give a number for the decrease and point out the significance level, The effect of the mutant is so small that this is certainly not the main /direct regulator involved. Moreover, the results should not be hidden in the supplement

R: Thanks for your comment. The intensity of the fluorescence was decreased about 31% in the TF *guaR* (LPD06577) deletion mutant when compared to the WT strain. The change of the fluorescence is significant (for statistics analysis, please see Fig. 2f), but the benzoate consumption and cell growth data indicated that this TF is not the direct regulator responsible for the regulation of BEN degradation. In the revised manuscript, we showed these data in new Fig. 2.

270 identities of 26% mean that the proteins are totally different from each other. Therefore, it is clear that these mutants also had no effect. The mentioned "significant decrease" is very questionable, since the about same reporter activity in all mutants indicates that the decrease is caused by some technical difference between the WT and mutant experiments (presence/absence of antibiotics or anything else??).

R: Thanks for your comment. We agree with you that the low identities of these TFs may suggest that they are totally different regulators. We just removed it from our revised manuscript.

276 what is meant with "significant induction is found" in the triple mutant??? I see (almost) no difference between WT and any of the mutants.

R: Thanks for your comment. What we want to describe here is that even in the triple mutant, we still see the benzoate funneling pathway can be induced when treated with benzoate. We have removed this part from our revised manuscript.

286 the lengthy description of how the reporter was made belongs into Methods (moreover, it is the same procedure as before once again)

R: Thanks for your comment and suggestion. The details of how we generated the *gfp*⁺ transcriptional constructs were moved into the Methods section.

295 why all the jumping between the figures (supp. Fig 1e here, but supp. Fig. 3 in the previous chapter)

R: We just have all the data of selective responses of aromatic responsive promoters in Supplementary Figure 1. So, when we talk about different promoters, it will jump.

326 ff I am totally giving to follow what the authors did with the "funneling pathways". The way the experiments and results are described, I can not figure out what the authors did and how to judge the results. The paper needs to be completely re-written with reasonable gene designations and separation of methods and results.

R: Thanks for your comments and suggestions. We have substantially revised our manuscript, and hope the quality has been improved.

Discussion: much too long (& pages) and mostly repeats what has been stated in Results.
Methods: almost nothing is in there. A lot of superfluous stuff from the Results should be described here.

R: Thanks for your suggestion. We have rewritten the Discussion section, and more details of the experiments have been added in Methods, including how we generated the plasmids in this study, how we did the CRISPRi experiment, and how we prepares the seed culture for the carbon catabolite repression study.

Reviewer #2 (Remarks to the Author):

The authors have used gene knockout strategies to identify and assign activator and repressor roles to transcription factors that enable a better understanding of transcriptional regulation in Rhodococcus opacus relevant to aromatic degradation pathways for lignin valorisation. The work is described well and can be seen as a valuable addition to the efforts of developing synthetic biology strategies for lignocellulosic valorisation efforts and beyond. Whilst the work appears to be sufficiently novel to merit publication, the generic title of the manuscript is a bit misleading. I would urge the authors to think of a more meaningful title that specifically reflects the work carried out, including the name of the organism they have worked with. It is advisable to have a title that reflects the key specific findings of the research. It would also help for the authors to state clearly the specific findings and how this advances the field. The summarised findings in the abstract come across as being a bit too general, lacking in specifics. Perhaps, it would help if the authors could be more specific about the novel findings of the investigation in the abstract and the title. Otherwise, I believe the authors have done a good job.

R: Thanks for your evaluation of our work. We have changed the title and revised the introduction/abstract with more specifics included.

REVIEWERS' COMMENTS:

Reviewer #1 (Remarks to the Author):

I approve of the changes made by the authors.